# A fusion peptide in preS1 and the human protein disulfide isomerase ERp57 are involved in hepatitis B virus membrane fusion process

Jimena Pérez-Vargas[1†], Elin Teppa[2,3†], Fouzia Amirache[1], Bertrand Boson[1], Rémi Pereira de Oliveira[1], Christophe Combet[4], Anja Böckmann[5], Floriane Fusil[1], Natalia Freitas[1‡], Alessandra Carbone[2‡*], François-Loïc Cosset[1‡*]

[1]CIRI – Centre International de Recherche en Infectiologie, Univ Lyon, Université Claude Bernard Lyon 1, Inserm, U1111, CNRS, UMR5308, ENS Lyon, Lyon, France; [2]Sorbonne Université, CNRS, IBPS, Laboratoire de Biologie Computationnelle et Quantitative (LCQB) - UMR 7238, Paris, France; [3]Sorbonne Université, Institut des Sciences du Calcul et des Données (ISCD), Paris, France; [4]Cancer Research Center of Lyon (CRCL), UMR Inserm 1052 - CNRS 5286 - Université Lyon 1 - Centre Léon Bérard, Lyon, France; [5]Molecular Microbiology and Structural Biochemistry, UMR5086 CNRS-Université Lyon 1, Lyon, France

*For correspondence:
Alessandra.Carbone@lip6.fr (AC);
Francois-Loic.Cosset@ens-lyon.fr (F-LC)

†These authors contributed equally to this work
‡These authors also contributed equally to this work

**Competing interests:** The authors declare that no competing interests exist.

**Abstract** Cell entry of enveloped viruses relies on the fusion between the viral and plasma or endosomal membranes, through a mechanism that is triggered by a cellular signal. Here we used a combination of computational and experimental approaches to unravel the main determinants of hepatitis B virus (HBV) membrane fusion process. We discovered that ERp57 is a host factor critically involved in triggering HBV fusion and infection. Then, through modeling approaches, we uncovered a putative allosteric cross-strand disulfide (CSD) bond in the HBV S glycoprotein and we demonstrate that its stabilization could prevent membrane fusion. Finally, we identified and characterized a potential fusion peptide in the preS1 domain of the HBV L glycoprotein. These results underscore a membrane fusion mechanism that could be triggered by ERp57, allowing a thiol/disulfide exchange reaction to occur and regulate isomerization of a critical CSD, which ultimately leads to the exposition of the fusion peptide.

## Introduction

Hepatitis B is a major public health problem; it affects over 250 million people worldwide, and 850,000 deaths occur each year as a result of hepatitis B complications. The structure of its etiological agent, the hepatitis B virus (HBV), features a nucleocapsid that is surrounded by a lipid bilayer containing the HBV surface antigen (HBsAg) consiting in three envelope glycoproteins (GPs) designated as small (S), medium (M), and large (L), which are the products of a single open reading frame. They share the C-terminal S domain that contains four putative transmembrane (TM) domains. The L and M proteins have N-terminal extensions (preS1/preS2 and preS2, respectively) that mediate diverse functions in nucleocapsid binding and receptor recognition (*Baumert et al., 2014*). The first 2- to 75-amino-acid sequence of the preS1 domain of the L protein (*Blanchet and Sureau, 2007*; *Bremer et al., 2011*; *Le Seyec et al., 1999*) and the antigenic loop (AGL) of the S domain (*Le Duff et al., 2009*; *Salisse and Sureau, 2009*; *Schulze et al., 2007*) have been identified as essential determinants for infectivity of HBV and hepatitis delta virus (HDV), a pathogen that depends on HBV GPs for its propagation.

Entry of enveloped viruses into cells can be defined as the sequence of events occurring from the attachment of the virus to the host cell until the release of the genome into the cytoplasm, via fusion between viral and cellular membranes. Like for most enveloped viruses, HBV entry into cells is a finely regulated and complex process consisting of different steps, in which several viral and cellular factors are involved. Its first step involves low-affinity binding to heparan sulfate proteoglycans (HSPGs) residing on the hepatocytes' surface (*Leistner et al., 2008*; *Schulze et al., 2007*). This attachment is mediated by the preS1 region of the L protein and/or the AGL of the S protein (*Ni et al., 2014*; *Schulze et al., 2007*). Afterwards, the virus interacts with its high-affinity receptor, the sodium taurocholate-cotransporting polypeptide (NTCP) (*Ni et al., 2014*; *Yan et al., 2012*), through the amino-terminal end of the L protein preS1 domain (*Glebe et al., 2005*; *Gripon et al., 2005*; *Yan et al., 2012*). NTCP is an integral membrane protein expressed at the basolateral membrane of hepatocytes, which explains the tropism of HBV for the liver.

The post-binding entry steps of HBV occur through endocytosis; however, the exact mechanism is still unclear and somehow controversial. One early study in HepaRG cells showed that HBV is internalized via caveolin-mediated endocytosis (*Macovei et al., 2010*). Nevertheless, inhibition of caveolin-mediated endocytosis or silencing of caveolin-1 did not impair HBV infection in tupaia hepatocytes (*Bremer et al., 2009*) or in HepaG2-NTCP cells (*Herrscher et al., 2020*). Contrastingly, several other studies presented evidence that HBV endocytosis is clathrin-dependent (*Herrscher et al., 2020*; *Huang et al., 2012*; *Umetsu et al., 2018*). Recent studies have reported that HBV infection of HepaRG cells depends on Rab5 and Rab7 (*Macovei et al., 2013*), which are GTPases involved in the biogenesis of endosomes, and that the epidermal growth factor receptor (EGFR) is a host-entry cofactor that interacts with NTCP and mediates HBV internalization (*Iwamoto et al., 2019*). These findings support the hypothesis that HBV is transported from early to mature endosomes. After the early endosome stage, translocation is associated with a gradually decreasing pH, from about 6.2 in early endosomes to close to 5.5 in late endosomes, which allows fusion of many enveloped viruses with the endosomal membrane. However, in the case of HBV, pharmacological agents that raise or neutralize the pH in the endocytic pathway do not affect infection (*Macovei et al., 2010*; *Macovei et al., 2013*; *Rigg and Schaller, 1992*). Furthermore, treatments with protease inhibitors have no effect on infection (*Macovei et al., 2013*), suggesting that HBV transport into the degradative branch of the endocytic pathway is not required per se to initiate this process.

Virus entry by membrane fusion involves interactions between viral fusion proteins and host receptors, which result in conformational changes of the virus envelope proteins. However, the molecular determinants and mechanism of membrane fusion of HBV remain to be defined. Previous results have indicated the essential role of the cysteine residues of the AGL, as shown by the reduction of virus entry levels by inhibitors of thiol/disulfide exchange reaction (*Abou-Jaoudé and Sureau, 2007*), hence suggesting a redox state responsible for conformational changes that can have a role during the fusion step.

Here, using a combination of computational and experimental approaches, we sought to better understand how HBV induces the fusion of its lipid membrane with that of the infected cell. Specifically, using a coevolution analysis of HBV GPs and molecular modeling combined with experimental investigations ex vivo in molecular virology and in vivo in liver humanized mice, we provide evidence that the mechanism triggering HBV membrane fusion involves ERp57, a cellular protein disulfide isomerase (PDI). Furthermore, our results highlight the role of specific cysteines in the AGL determinant as well as a sequence (aa 48–66) in the preS1 determinant that could ultimately act as a fusion peptide mediating HBV membrane fusion.

## Results

### HBV membrane fusion is independent of acidic pH and receptor expression

To investigate the fusion activation mechanism and to identify the fusion determinants of HBV, we designed a cell-cell fusion assay whereby Huh7 'donor' cells, expressing a luciferase reporter gene under control of the HIV-1 promoter, were co-cultured with either Huh7-tat or Huh7-NTCP-tat 'indicator' cells, expressing the HIV-1 transactivator of transcription (Tat) protein, which induces luciferase

expression only in fused donor and indicator cells (*Lavillette et al., 2007*). We transfected donor cells with pT7HB2.7 (*Sureau et al., 1994*), an expression plasmid encoding the wild-type HBV glycoproteins L, M, and S. The transfected donor cells were then co-cultivated with Huh7-tat or Huh7-NTCP-tat indicator cells for 1 day. The medium of the co-cultures was then acidified at pH 4 for 3 min to trigger fusion and the next day, the luciferase activity in the lysates of co-cultured cells was measured as a read-out of membrane fusion (*Figure 1A*). The GPs of vesicular stomatitis virus (VSV) or of Crimean-Congo hemorrhagic fever virus (CCHFV) were used as controls for viruses that need acidic pH to promote membrane fusion. We found that HBV GPs induced similar levels of fusion in co-cultures that were exposed to either acidic or neutral pH, as well as in co-cultures lacking or expressing the NTCP receptor (*Figure 1A*; see raw data in *Figure 1—figure supplement 1*). Since HBV entry requires HSPGs to mediate the capture of its viral particles through HBsAg (*Leistner et al., 2008*; *Schulze et al., 2007*), we addressed whether blocking of HBsAg/HSPG interaction could inhibit cell-cell fusion using heparin as the competitor. Yet, while the applied doses of heparin could prevent cell-free entry, as shown previously (*Schulze et al., 2007*), addition of soluble heparin to the co-cultures did not prevent HBsAg-mediated fusion, whether the indicator cells expressed NTCP or not (*Figure 1B*). We confirmed these results by using CHO and CHO-pgsB618 (*Richard et al., 1995*) cells as donor and/or indicator cells. While both cell types do not express NTCP, only the former expresses HSPGs. We found that cell-cell fusion could be detected for either indicator cell type to the same extent as for Huh7 cells (*Figure 1C*).

Altogether, these results indicated that cell-cell fusion mediated by HBV GPs is independent of acidic pH and requires neither HSPG nor NTCP receptor, which underscores an alternative fusion trigger.

## The preS1 domain of HBV L protein harbors a critical determinant of membrane fusion

The L, M, and S GPs of HBV are produced by a single open reading frame and share a common C-terminal S domain. M and L proteins harbor additional N-terminal extensions (preS2 and preS1/preS2, respectively), with preS1 harboring the NTCP-binding determinant (*Glebe et al., 2005*; *Gripon et al., 2005*). Noteworthy, the fusion determinants of HBV GPs and, particularly, the fusion peptide that could induce merging of viral and endosomal membranes, have not yet been functionally identified in infection or cell-cell fusion assays.

First, to address which GP is responsible for HBV membrane fusion, we evaluated the role of either protein in cell-cell fusion assays (*Figure 1D*). Huh7 cells were transfected with plasmids encoding wild-type (wt) HBV GPs, that is, L, M, and S (pT7HB2.7 plasmid) vs only L, M, or S (using pCiL, pCiM, and pCiS plasmids, respectively) (*Komla-Soukha and Sureau, 2006*). To analyze the expression of either protein at cell surface, transfected cells were labeled with sulfo-NHS-SS-biotin, a chemical compound that is unable to penetrate biological membranes. After lysis and immunoprecipitation of biotinylated proteins, we found that the individually expressed L, M, or S proteins were detected at similar levels as compared to HBV GPs (L, M, and S) expressed simultaneously, as in cells transfected with the wt pT7HB2.7 plasmid (*Figure 1E and F*). Then, to determine the fusion activity of either protein, we performed cell-cell fusion assays as described above. We found that none of the L, M, or S proteins expressed alone were able to induce membrane fusion (*Figure 1D*). Furthermore, when we tested the pT7HB2.7Mless plasmid, which induces co-expression of S and L only ('noM' in *Figure 1D–F*), we detected a cell-cell fusion activity at the same level than for wt HBV GPs (*Figure 1D*). This indicated that M is not necessary for membrane fusion, in agreement with previous results (*Ni et al., 2010*; *Sureau et al., 1994*) showing that M is dispensable for infectivity of viral particles (*Figure 1—figure supplement 3*).

Altogether, these results suggested that the determinants of membrane fusion are harbored within L and S GPs.

Next, aiming to identify a fusion peptide in either protein, we used a computational approach to pinpoint regions of the HBV GPs that may potentially interact with membrane bilayers. Using Membrane Protein Explorer (MPEx), a tool based on the Wimley-White interfacial hydrophobicity scale (*Snider et al., 2009*), five regions of high interfacial hydrophobicity were identified (*Figure 2—figure supplement 1A*). Two out of the five hydrophobic regions did not correspond to HBV GP transmembrane regions (TM1, TM2, and TM3/TM4) and, therefore, were considered as candidate fusion peptides (*Figure 2A and B*). The first predicted segment comprised amino acids 48–66 that overlap

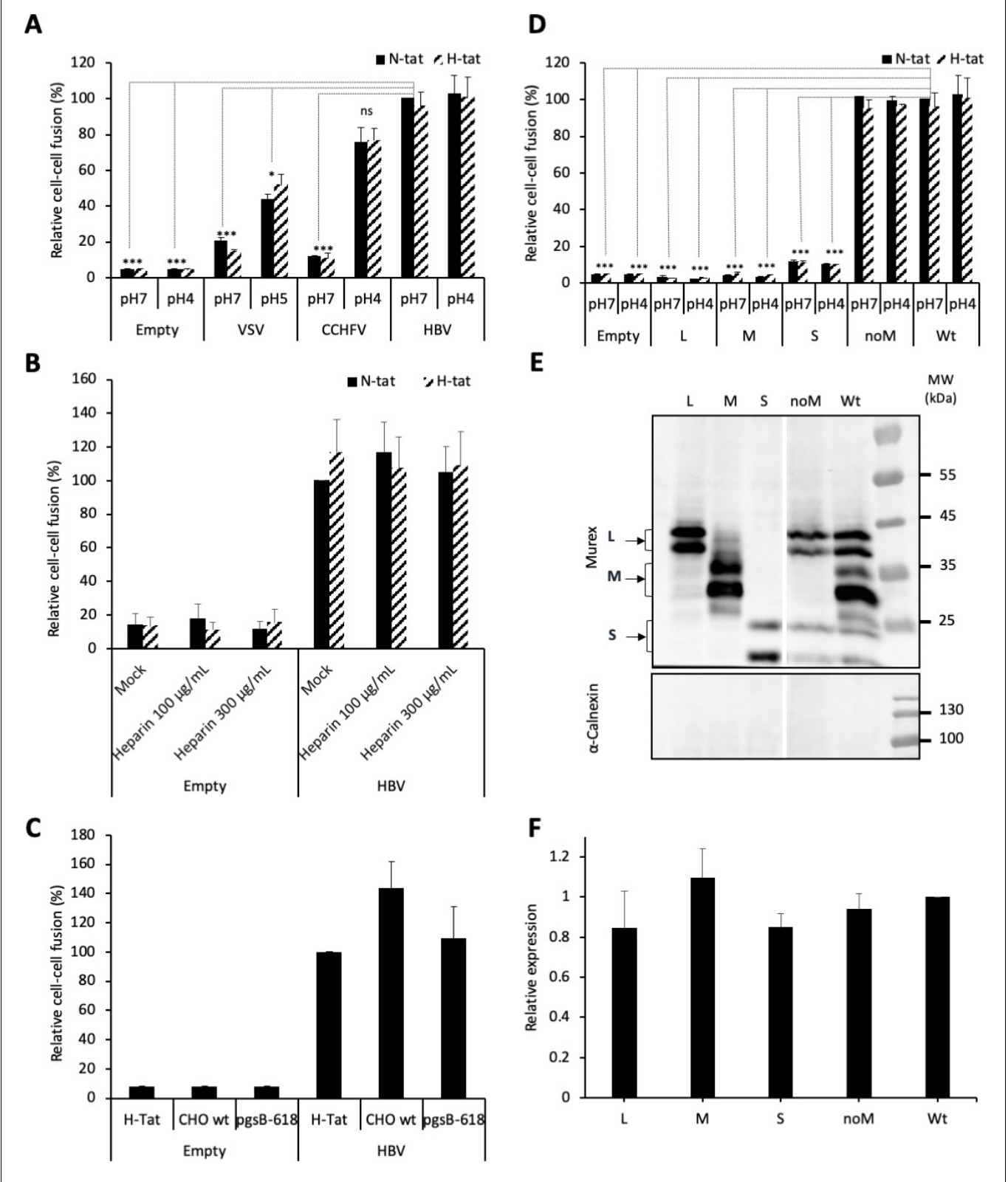

**Figure 1.** HBV GP fusion trigger is independent of acidic pH, HSPG, and NTCP. (**A**) Huh7 'donor' cells transfected with the pT7HB2.7 plasmid allowing expression of hepatitis B virus glycoproteins (HBV GPs) (HBV) and a luciferase marker gene driven by the HIV-1 promoter were co-cultured with either Huh7-tat (H-tat) or Huh7-NTCP-tat (N-tat) 'indicator' cells that express the HIV Tat protein. After 24 hr of co-culture, the cells were treated at pH 4 (or pH 5 for VSV-G) vs pH 7 for 3 min. The luciferase activity induced by fusion between donor and indicator cells was then measured 24 hr later. A control

*Figure 1 continued on next page*

Figure 1 continued

plasmid that does not allow GP expression (Empty) was used to determine the background of luciferase expression. The Crimean-Congo hemorrhagic fever virus (CCHFV) Gn/Gc (CCHFV) or vesicular stomatitis virus-G (VSV-G) (VSV) GPs were used as positive controls for fusion at low pH. Fusion mediated by HBV GPs with Huh7-tat cells was taken as 100%. The bars represent the means (N = 3). Error bars correspond to standard deviation. See the raw data of individual experiments in *Figure 1—figure supplement 1*. (B) Results of cell-cell fusion assays performed as described above in the presence of heparin at the indicated concentrations throughout the co-culture. No cytotoxicity could be detected in these conditions (*Figure 1—figure supplement 2*). The graphs represent the average of two independent experiments. Fusion mediated by HBV GPs with mock-treated Huh7 cells was taken as 100%. (C) CHO 'donor' cells transfected with the pT7HB2.7 plasmid and a luciferase marker gene driven by the HIV-1 promoter were co-cultured with either Huh7-tat (H-tat), CHO-tat (CHO wild-type [wt]), or CHO-pgsB618-tat (pgsB618) 'indicator' cells that express the HIV Tat protein. The luciferase activity induced by fusion between donor and indicator cells was then measured 24 hr later. A control plasmid that does not allow GP expression (Empty) was used to determine the background of luciferase expression. Fusion mediated by HBV GPs with Huh7-tat was taken as 100%. The graphs represent the average of two independent experiments. (D) Huh7 'donor' cells transfected with plasmids allowing expression of L, M, or S HBV GPs alone, both L and S GPs (noM), or all HBV GPs (Wt) and a luciferase marker gene driven by the HIV-1 promoter were co-cultured with Huh7-tat or Huh7-NTCP-tat 'indicator' cells that express HIV Tat protein. Cell co-cultures were then processed as described above to determine cell-cell fusion activity. Fusion mediated by HBV GP at pH 7 with Huh7-tat cells was taken as 100%. The bars represent the means (N = 3). Error bars correspond to standard deviation. (E) Detection of HBV GPs at the cell surface by biotinylation. Transfected Huh7 cells were biotinylated for 30 min at 4°C and then processed biochemically. Cell lysates were subjected to streptavidin pull-down prior to western blot analysis using anti-HBsAg antibody (Murex). The molecular weight markers (kDa) are shown on the right. Calnexin detection was used as control for the cytoplasmic protein marker, showing the integrity of cell membrane, as shown in this representative western blot. (F) Relative GP expression at the cell surface as compared to Wt, quantified by adding the L+M+S signals from western blot analyses. The results are expressed as mean ± SD (N = 3). No statistical differences could be found using the Mann-Whitney test (p-value>0.05). See also the quantification of total HBV GP expression in *Figure 1—figure supplement 4*.

The online version of this article includes the following source data and figure supplement(s) for figure 1:

**Source data 1.** HBV GP fusion trigger is independent of acidic pH and NTCP.
**Source data 2.** HBV GP fusion trigger is independent of acidic pH and NTCP.
**Figure supplement 1.** HBV GP fusion trigger is independent of acidic pH and NTCP.
**Figure supplement 2.** Results of cell survival after drug treatments.
**Figure supplement 3.** Characterization of 'noM' HDV particles.
**Figure supplement 4.** Total protein expression.

with the preS1 domain. The second segment, which includes amino acids 127–145, is included in the preS2 region. Our prediction analyses indicated that the first segment (ΔG = -3.38) was more likely to be a fusion peptide than the second one (ΔG = -0.85) (*Figure 2—figure supplement 1B*). Considering the Wimley-White scale, a set of mutants was designed to alter the hydrophobicity of the two predicted segments (*Figure 2B* and *Figure 2—figure supplement 1B*). In the first segment, three mutants were studied by changing the aromatic residues to an alanine or glutamate: F52A, F56A, W66A, F52A/W66A (FW/AA), and F52E/W66E (FW/EE), or a glycine to an alanine (G53A). In the second segment, four mutants were considered: Y129A, F130A, S136E, and L144A; while the first two mutants targeted aromatic residues, S136 and L144 were also considered important because they are at the center of the predicted region and have a relatively high hydrophobicity.

To evaluate the role of these two sequences in HBV fusion, we introduced these single or double mutations in both regions and inserted them in the pT7HB2.7 HBV GP expression plasmid. Each mutant was compared to wt HBV GPs in both infection assays, using HDV particles (*Sureau, 2010*; *Perez-Vargas et al., 2019*), and cell-cell fusion assays, as described above. We found that HDV particles carrying these mutant GPs were produced by Huh7 cells at levels similar to those produced with wt GPs (*Figure 2C and D*), hence ruling out gross misfolding induced by the mutations that would otherwise prevent HBV GP incorporation into viral particles (*Abou-Jaoudé and Sureau, 2007*). Interestingly, no infectivity could be detected for most of the mutations introduced in the preS1 peptide (*Figure 2E*), whereas the HDV particles with mutations in the preS2 peptide showed levels of infectivity that were similar to those obtained with the wt GPs (*Figure 2F*). Correlating with the results of these infection assays, we found that the mutants in the preS1 peptide that prevented HDV infectivity also abrogated cell-cell fusion activity (*Figure 2G*) in a manner unrelated to the levels of GP cell-surface expression (*Figure 2I* and *Figure 2—figure supplement 2*). In contrast, mutations in the preS2 peptide displayed the same levels of cell-cell fusion activity as compared to wt (*Figure 2H and J*).

Altogether, these results indicated that the preS1 region harbors a potential fusion peptide.

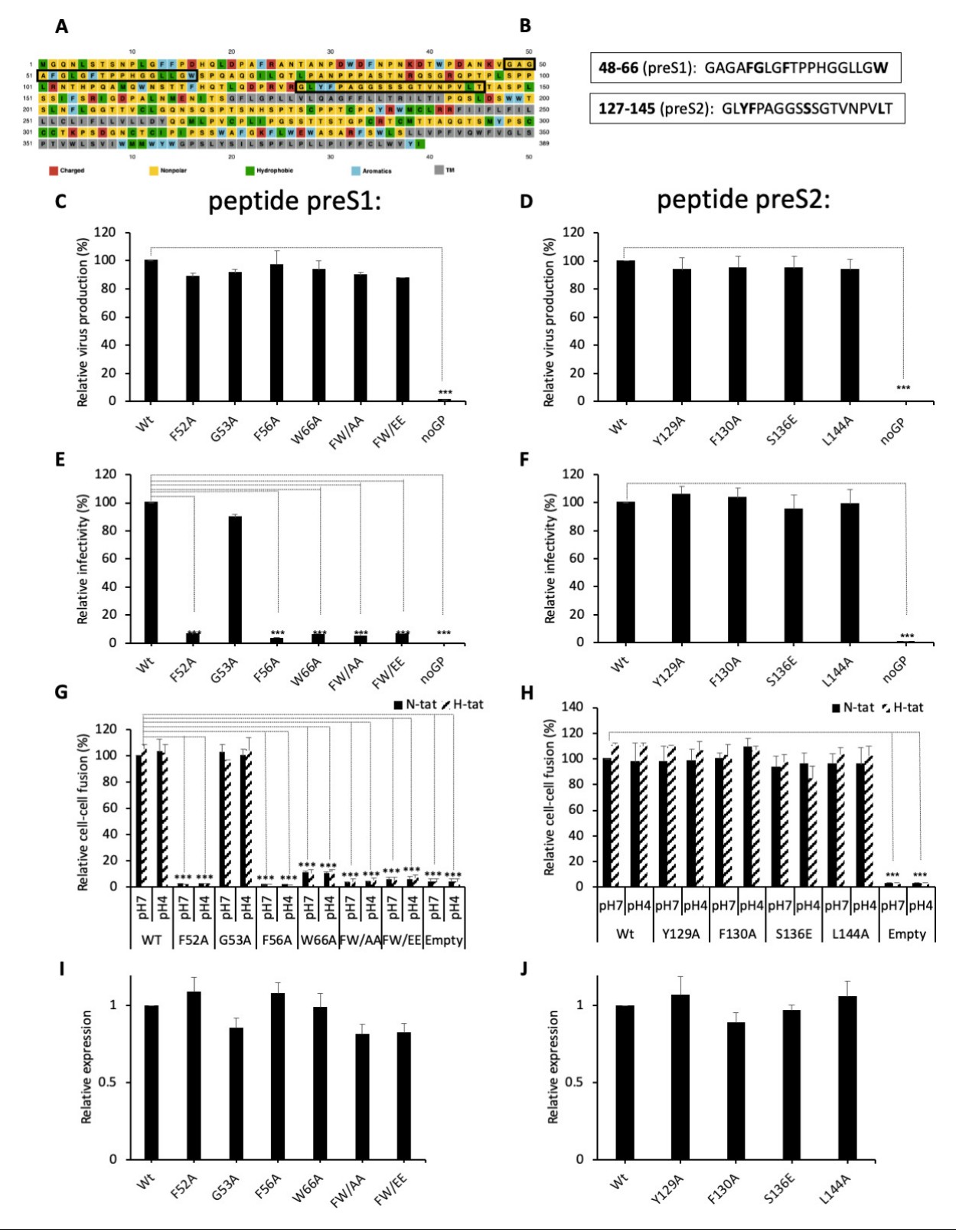

**Figure 2.** Functional analysis of predicted HBV fusion peptides. (**A**) Sequence of hepatitis B virus (HBV) L protein showing the amino acid color code and boxes for the localization of the two predicted fusion peptides in preS1 and in preS2. (**B**) Sequences of the two predicted fusion peptides showing the positions that were mutated (bold). (**C, D**) Huh7 cells were co-transfected with pSVLD3 plasmid coding for hepatitis delta virus (HDV) RNPs and plasmids coding for wild-type (wt) or mutant HBV glycoproteins (GPs). The FW/AA and FW/EE are double-alanine mutants at positions F52 and W66.

*Figure 2 continued on next page*

*Figure 2 continued*

As control, pSVLD3 was co-transfected with an empty plasmid (referred to as 'noGP'). At day 9 post-transfection, the cell supernatants were harvested and filtered, and the extracellular RNA was extracted and purified before quantifying HDV RNAs by quantitative reverse transcription PCR (RTqPCR). HDV RNA levels in GE (genome equivalent) are expressed as means ± SD (N = 3) per ml of cell supernatants. (E, F) HDV particles were used to infect Huh7-NTCP cells, which were grown for 7 days before total intracellular RNA was purified. The results of HDV RNA quantification by RTqPCR are expressed after normalization with glyceraldehyde 3-phosphate dehydrogenase (GAPDH) RNAs as means ± SD (N = 3) per ml of cell lysates containing $10^6$ cells. (G, H) Huh7 'donor' cells co-expressing wt or mutant HBV GPs and a luciferase marker gene driven by the HIV-1 promoter were co-cultured with either Huh7-tat (H-tat) or Huh7-NTCP-tat (N-tat) 'indicator' cells that express HIV Tat protein. After 24 hr, the cells were treated at pH 4 or pH 7 for 3 min. The luciferase activity induced by the fusion between the donor and indicator cells was measured 24 hr later. Fusion mediated by wt GP at pH 7 with Huh7-NTCP-tat cells was taken as 100%. The bars represent the means (N = 5). Error bars correspond to standard deviations. (I, J) Quantification of wt and mutant GPs at cell surface by western blot analyses (see examples in *Figure 2—figure supplement 2*). The results show the relative GP expression of preS1 (I) and preS2 (J) mutants compared to Wt, as indicated, and are expressed as means ± SD (N = 3). No statistical differences could be found using the Mann-Whitney test (p-value>0.05).

The online version of this article includes the following source data and figure supplement(s) for figure 2:

**Source data 1.** Functional analysis of predicted HBV fusion peptides.

**Source data 2.** Functional analysis of predicted HBV fusion peptides.

**Source data 3.** Functional analysis of predicted HBV fusion peptides.

**Figure supplement 1.** Prediction of fusion peptides within S protein by using Wimley-White interfacial hydrophobicity scale.

**Figure supplement 2.** Cell-surface and intracellular detection of preS1 and preS2 HBV GP mutants.

## Stabilizing cross-strand disulfide exchanges in HBV S protein prevents membrane fusion

Next, we sought to investigate the mechanisms that could induce fusion-activating conformational changes in the HBV GPs, leading to exposure of the fusion peptide. As neither the HBV receptor interaction nor the acidic pH could trigger membrane fusion (*Figure 1*), we thought that conformational rearrangement of HBV GPs might involve reshuffling of their disulfide bonds. Indeed, previous studies have shown that cysteine residues of the HBV S antigenic loop are essential for HDV infectivity and that viral entry is blocked by inhibitors of thiol/disulfide exchange reactions, such as Tris(2-carboxyethyl)phosphine hydrochloride (TCEP), dithiothreitol (DTT), 5,5-dithiobis(2-nitrobenzoic acid) (DTNB), or 4-acetamido-4'-maleimidyl-stilbene-2,2'-disulfonate (AMS) (*Abou-Jaoudé and Sureau, 2007*). Thus, to extend the notion that thiol/disulfide exchange reactions are implicated during membrane fusion and entry, we performed HBV infection and fusion assays in the presence of DTNB, an alkylating agent. First, using different DTNB concentrations that were added either at the onset of infection or at 16 hr post-infection, we confirmed that DTNB could block HDV infection in a dose-dependent manner, but only when it was added at the onset of infection (*Abou-Jaoudé and Sureau, 2007*; *Figure 3—figure supplement 1*). Second, using time-of-addition experiments, we found that DTNB could inhibit infection only if added within the first 2 hr after inoculation with HDV particles (*Figure 3A*). These results suggested that DTNB blocks a thiol/disulfide exchange reaction that could be necessary at an early step of infection, such as a trigger of the fusion mechanism, though not at a later stage of the entry process. Third, to evaluate the effect of DTNB on membrane fusion, we performed cell-cell fusion assays in presence of DTNB, which was added at the onset of cell co-cultures vs at 16 hr after seeding the cell co-cultures. We showed that DTNB added during the co-culture neither induced cytotoxicity (*Figure 1—figure supplement 2*) nor affected expression of HBV glycoproteins on the cell surface (*Figure 3C and D*). Yet, we found a dose-dependent reduction in the level of cell-cell fusion when DTNB was added immediately after cell-cell contact, whereas we detected a much lower effect in fusion activity when DTNB was added at 16 hr after cell contact (*Figure 3B*).

Altogether, these results suggested a role of the disulfide bond network during HBV membrane fusion steps, perhaps at the level of the fusion trigger.

To address this possibility and to identify potential mechanisms involved in fusion triggering, we focused on the 'a' determinant of protein S that exhibits eight conserved Cys, which, for some of them, are in strong proximity in the sequence (*Figure 4A*). To avoid trivial contact predictions between consecutive Cys, we defined four Cys-containing regions in a way that Cys pairs that are potentially in contact should have a sequence separation of at least four amino acids. The first Cys-containing region includes C270, the second, C284 and C287, the third, C300, C301, and C302, and

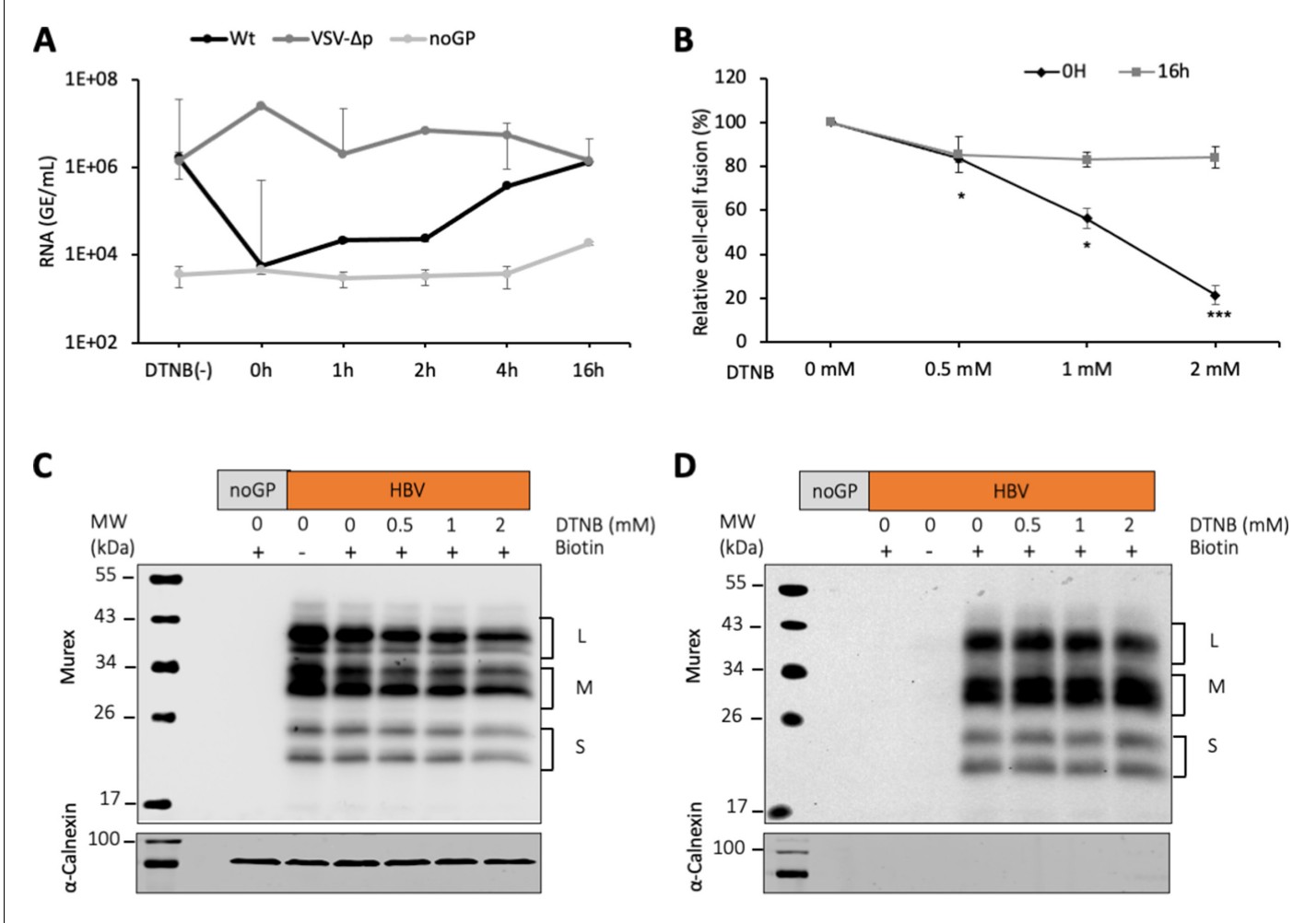

**Figure 3.** DTNB, a thiol-specific oxidizing reagent, inhibits HBV membrane fusion. (**A**) 5,5-Dithiobis(2-nitrobenzoic acid) (DTNB) (2 mM) was added to the cell supernatant containing hepatitis delta virus (HDV) particles at the onset of infection (0 hr) or at the indicated times post-infection and was removed 8 hr later. VSV-Δp, that is, HDV particles generated with vesicular stomatitis virus-G glycoprotein (VSV-G GP) rather than hepatitis B virus (HBV), were used as the control for a virus entry process that is not affected by DNTB. As a negative control, pSVLD3 was co-transfected with an empty plasmid (referred to as 'noGP'). At 7 days post-infection, HDV RNAs were extracted from infected cells and quantified by quantitative reverse transcription PCR (RTqPCR). The results are expressed after normalization with glyceraldehyde 3-phosphate dehydrogenase (GAPDH) RNAs as means ± SD (N = 3) per ml of cell lysates containing $10^6$ cells. The results of infection in the absence of DTNB are shown (DTNB(-)). (**B**) Huh7 'donor' cells co-expressing HBV GPs and a luciferase marker gene driven by the HIV-1 promoter were co-cultured with Huh7-NTCP-tat 'indicator' cells that express HIV Tat protein. Different concentrations of DTNB were added at 0 hr vs at 16 hr after initiating the cell co-culture, as indicated. No cytotoxicity could be detected in these conditions (*Figure 1—figure supplement 2*). The luciferase activity induced by fusion between donor and indicator cells was then measured 24 hr later. Fusion mediated by HBV GPs without DTNB was taken as 100%. The graphs represent the average of four independent experiments. (**C, D**) Huh7 cells transfected with pUC19 (noGP) or the pT7HB2.7 (HBV) plasmids were incubated with dimethyl sulfoxide (DMSO) (0) or increasing doses of DTNB (0.5, 1, and 2 mM) for 16 hr prior to incubation with biotin for 30 min at 4°C. Biotin was omitted from one sample (-) and served as a negative control for non-specific binding of proteins to streptavidin. Cells were subsequently lysed and the biotinylated surface proteins were captured by streptavidin agarose. Total (**C**) and biotin-labeled proteins (**D**) were then analyzed by western blot using anti-HBsAg (Murex) and anti-calnexin antibodies. Calnexin detection was used as a control for the cytoplasmic protein marker, showing the integrity of the cell membrane, as shown in these representative western blots. The molecular weight markers (kDa) are shown on the left.

The online version of this article includes the following source data and figure supplement(s) for figure 3:

**Source data 1.** DTNB, a thiol-specific oxidizing reagent, inhibits HBV membrane fusion.
**Source data 2.** DTNB, a thiol-specific oxidizing reagent, inhibits HBV membrane fusion.
**Source data 3.** DTNB, a thiol-specific oxidizing reagent, inhibits HBV membrane fusion.
**Figure supplement 1.** Effect of DTNB on HDV entry.

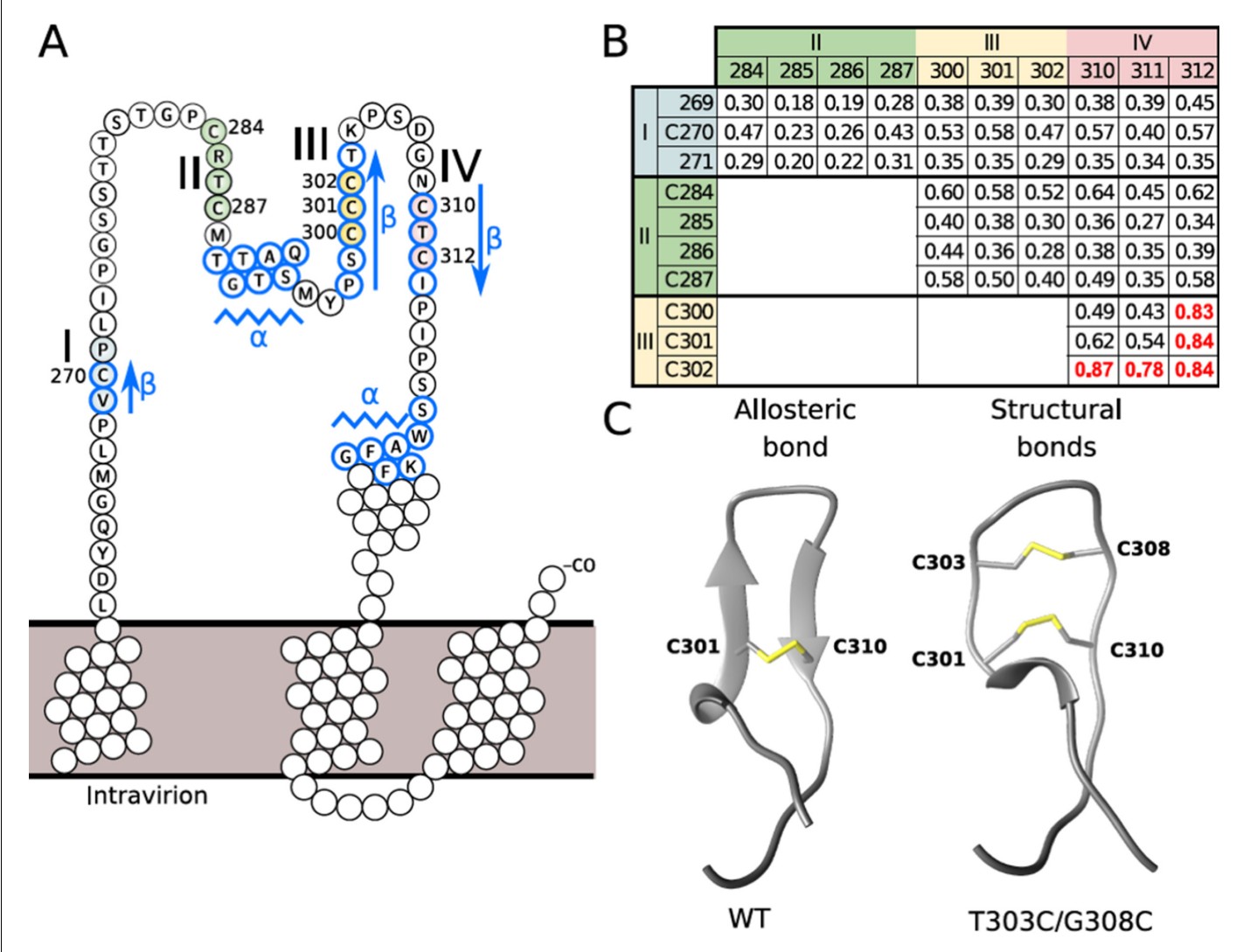

**Figure 4.** Disulfide conformation models. (A) Cysteine-rich regions on the 'a' determinant (residues 261–324) of the hepatitis B virus S glycoprotein (HBV S GP). Four subregions that are rich in cysteine are colored: I (blue), II (green), III (yellow), and IV (red). Jpred secondary structure prediction different from random-coil is indicated: β-strand (arrows) and α-helix (zigzag lines). (B) Probability of contacts predicted by RaptorX between the four cysteine-rich regions. The probabilities higher than 0.7 are highlighted in red (see also *Figure 4—figure supplement 1*). (C) Predominant disulfide conformations obtained by molecular dynamics simulation of the modeled 294–317 region of the HBV surface protein. Note that the ß-strand on the wild-type (wt) sequence (left) adopts a loop conformation with an allosteric disulfide conformer between the C301-C310 bond, which is specifically classified as a -/+RH Hook conformation. The T303C/G308C double mutant (right) may generate an additional disulfide bond, resulting in two structural disulfides of +/-RH Staple and -/+LH Spiral conformations that form the C301-C310 and C303-C308 bonds, respectively.

The online version of this article includes the following figure supplement(s) for figure 4:

**Figure supplement 1.** Contact map prediction for the L protein by RaptorX.

**Figure supplement 2.** Geometry of a disulfide bond.

the last one, C310 and C312 (*Figure 4A*). We applied secondary and tertiary structure prediction methods together with the contact prediction method RaptorX (*Ma et al., 2015*; *Wang et al., 2017*), based on coevolution signals, to predict disulfide connectivity in the 'a' determinant, which may identify which Cys forms disulfide bonds. Notably, RaptorX predicted structural contacts between either region (*Figure 4—figure supplement 1*) and we highlighted pairs of residues in contact in the four Cys regions, with the strongest signal detected between the third and fourth regions (*Figure 4B*). Next, applying the JPred secondary structure prediction method (*Cole et al., 2008*), we predicted two β-strands in the Cys-rich regions delimited by the S segments 298–303 and 310–313

(*Figure 4A*). Then, considering the secondary structure prediction and the contact prediction, we built a three-dimensional model for the region 294–317 (*Figure 4C*), which indicated that this sequence is compatible with a β-hairpin structural motif containing a cross-strand disulfide (CSD) bond between C301 and C310. Finally, through the analysis of its five χ dihedral angles (*Figure 4—figure supplement 2*), this disulfide bond was classified in a '-H Staple conformation', which is a particular type of disulfide geometry associated with allosteric functions by triggering a conformational change upon switching between the reduced and oxidized states (*Chiu and Hogg, 2019*; *Hogg, 2003*).

We, therefore, hypothesized that the redox state of this disulfide may act as an allosteric switch that could contribute to control conformational rearrangements of the S protein. Thus, we used our structural model of the C301-C310 disulfide bond to design a mutant of S that could disrupt this hypothetical allosteric function, that is, the T303C/G308C double mutant that induces an additional C303-C308 disulfide bond (*Figure 4C*). Further molecular dynamic (MD) simulations (1000 frames per MD trajectory) carried out to differentiate between allosteric and structurally stabilizing disulfides, where the disulfides can be classified based on their angles (*Figure 4—figure supplement 2*), showed that the T303C/G308C mutant predominantly forms a structural disulfide bond.

Aiming to validate our prediction that an additional disulfide bond between the two β-strands could, by stabilizing the 298–313 β-hairpin motif, prevent membrane fusion from occurring, we produced HDV particles carrying the individual (T303C or G308C) and double (T303C/G308C) mutations in HBV GPs. By measuring HDV RNAs in cell supernatants, we found that all mutants could produce comparable levels of viral particles relative to wt virus (*Figure 5A*), suggesting absence of gross alterations of HBV GP conformation that would otherwise preclude virion assembly (*Abou-Jaoudé and Sureau, 2007*). Importantly, we found that while HDV particles generated with GPs harboring the individual mutations were as infectious as wt, those that were produced with the T303C/G308C double mutation (noted TG/CC in *Figure 5*) and the putative additional C303-C308 CSD bond were not infectious (*Figure 5B*). Moreover, we found that HDV particles harboring GPs with this T303C/G308C mutation had similar binding levels on Huh7 cells than those generated with wt GPs (*Figure 6A*), underscoring a post-binding defect. Then, to address this possibility, we performed cell-cell fusion assays with either HBV GP mutant, which was readily expressed at the cell surface (*Figure 5C*). We found that whereas the single mutations displayed similar fusion activity as compared to wt HBV GPs, the T303C/G308C double mutation completely prevented HBV GP-induced cell-cell fusion activity (*Figure 5D*).

Altogether, these results suggested that the putative C303-C308 additional disulfide bond stabilizing the loop containing the C301-C310 CSD bond inhibited HBV entry and fusion, perhaps by preventing conformational rearrangements of HBV GPs that are required for promoting membrane fusion.

## ERp57 is a protein disulfide isomerase that promotes HBV entry and infectivity in vivo

We reasoned that isomerization of the C301-C310 CSD (*Figure 4*) or of another CSD of the AGL determinant with allosteric functions could facilitate some conformational rearrangements required to promote membrane fusion. We, therefore, hypothesized that such an isomerization could be induced by a host factor from the PDI family, which are enzymes that can both reduce and oxidize disulfide bonds.

To address if PDIs are involved in HBV entry, we tested the effect of inhibitors (NTZ, EGCG, rutin, bacitracin, PX-12) that target different PDI species (PDIA1, ERp5, ERp57, TMX1) for their effect in cell entry of viral particles. First, through binding assays of viral particles to Huh7 or Huh7-NTCP cells performed in the presence of either inhibitor, we found that none of these inhibitors affected binding of HDV particles generated with either wt or T303C/G308C mutant GPs (*Figure 6A*). Second, using infection assays with Huh7-NTCP cells pre-incubated with either inhibitor, we found that HDV particles had strongly reduced infectivity in presence of nitazoxanide (NTZ) and (−)-epigallocatechin 3-gallate (EGCG) inhibitors that both target ERp57 (*Figure 6B*). Third, we confirmed these results using infection assays with authentic HBV particles (*Figure 6C*). Finally, to demonstrate that the inhibitors acted at the level of membrane fusion, we performed cell-cell fusion assays, as described above, whereby either inhibitor was added at the onset of co-cultures of HBV GP-expressing Huh7 donor and Huh7-NTCP-tat indicator cells and was kept throughout the assay period.

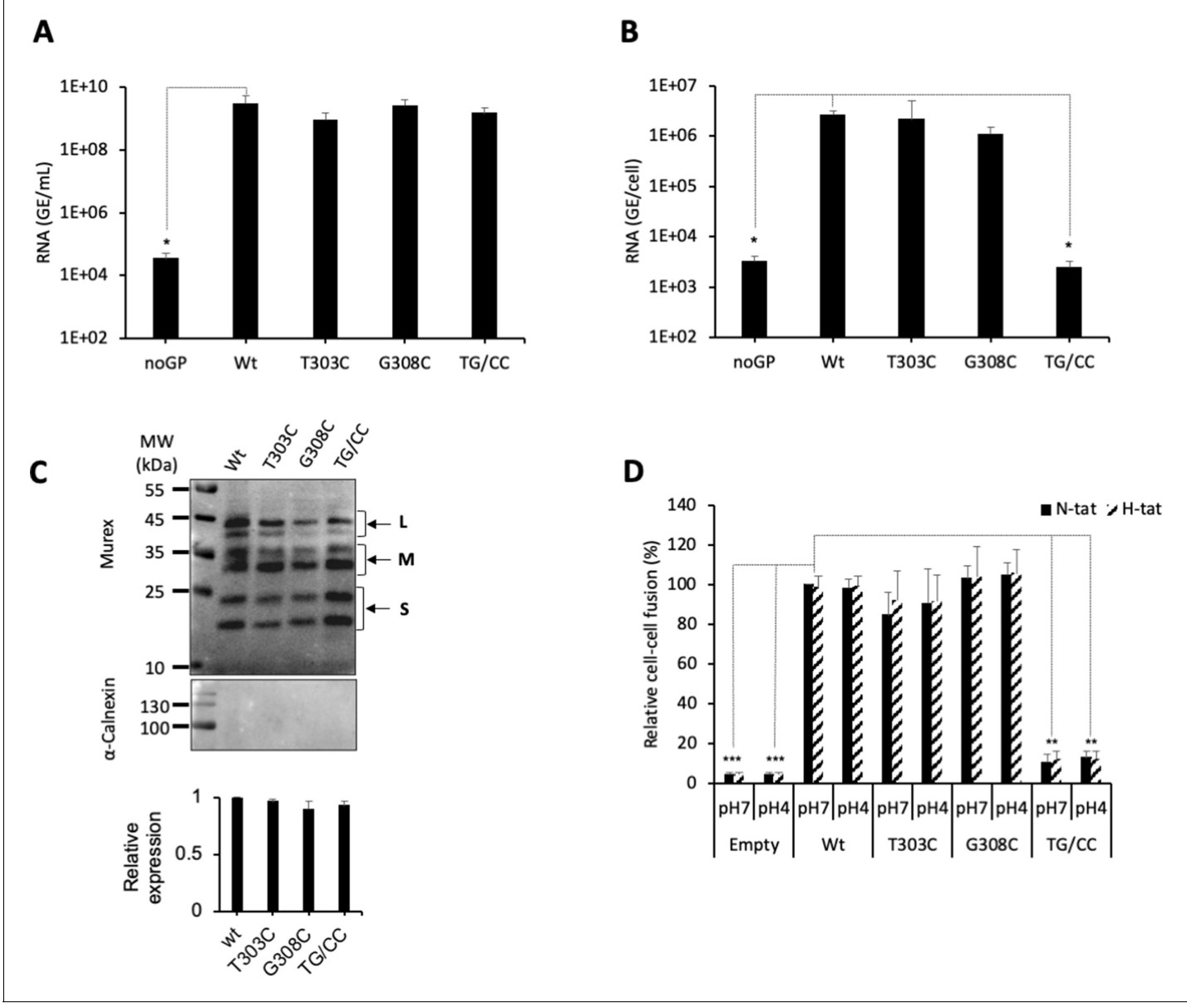

**Figure 5.** Evidence for a functional role of the CSD in the region 294–317 of the HBV S GP. (**A**) Huh7 cells were co-transfected with pSVLD3 plasmid coding for hepatitis delta virus (HDV) RNPs and plasmids coding for wild-type (wt), single, or double mutant (TG/CC) hepatitis B virus glycoproteins (HBV GPs). As control, pSVLD3 was co-transfected with an empty plasmid (referred to as 'noGP'). At day 9 post-transfection, the cell supernatants were harvested, filtered, and the extracellular RNA extracted and purified before quantifying HDV RNAs by quantitative reverse transcription PCR (RTqPCR). HDV RNA levels in GE (genome equivalent) are expressed as means ± SD (N = 4) per ml of cell supernatants. (**B**) HDV particles were used to infect Huh7-NTCP cells, which were grown for 7 days before total intracellular RNA was purified. The results of HDV RNA quantification by RTqPCR are expressed after normalization with glyceraldehyde 3-phosphate dehydrogenase (GAPDH) RNAs as means ± SD (N = 4) per ml of cell lysates containing $10^6$ cells. (**C**) Detection of GP mutants at the cell surface by biotinylation. Huh7 cells expressing wt or mutant HBV GPs were biotinylated for 30 min at 4°C and then processed biochemically. Cell lysates were subjected to streptavidin pull-down prior to western blot analysis using anti-HBsAg antibody (Murex). The molecular weight markers (kDa) are shown on the left. Calnexin detection was used as a control for the cytoplasm protein marker, showing the integrity of cell membrane, as shown in this representative western blot. The relative quantification of cell-surface GP expression compared to wt quantified from western blot analyses (means ± SD; N = 3) is shown below. See the quantification of total HBV GP expression in *Figure 1—figure supplement 4*. (**D**) Huh7 'donor' cells co-expressing wt or mutant HBV GPs and a luciferase marker gene driven by the HIV-1 promoter were co-cultured with either Huh7-tat (H-tat) or Huh7-NTCP-tat (N-tat) 'indicator' cells that express HIV Tat protein. After 24 hr, the cells were treated at pH 4 or pH 7 for 3 min. The luciferase activity induced by the fusion between the donor and indicator cells was measured 24 hr later. Fusion mediated by wt GP at pH 7 with Huh7-NTCP-tat cells was taken as 100%. The bars represent the means (N = 4). Error bars correspond to standard deviations.

*Figure 5 continued on next page*

*Figure 5 continued*

The online version of this article includes the following source data for figure 5:

**Source data 1.** Evidence for a functional role of the CSD in the region 294–317 of the HBV S GP.
**Source data 2.** Evidence for a functional role of the CSD in the region 294–317 of the HBV S GP.
**Source data 3.** Evidence for a functional role of the CSD in the region 294–317 of the HBV S GP.

Remarkably, we found a strong reduction in the levels of cell-cell fusion with the same drugs that inhibited HDV infection (*Figure 6D*). Hence, these results suggested a potential role of ERp57 in HBV membrane fusion.

Next, aiming to confirm and extend these findings, we selected a subset of the above PDIs, that is, ERp46, ERp57, and ERp72, which displayed low but significant expression at the surface of Huh7 cells (*Figure 7A*), in agreement with a previous report (*Turano et al., 2002*). We down-regulated either PDI in target cells via transduction of Huh7-NTCP cells with short hairpin RNA (shRNA)-expressing lentiviral vectors (*Figure 7—figure supplement 1* and *Figure 7—figure supplement 2*). We found that down-regulation of ERp57, though not ERp46 or ERp72, strongly reduced the levels of HDV (*Figure 7B*) and HBV infection (*Figure 7C*) and of cell-cell fusion (*Figure 7D*). Finally, through confocal microscopy analysis of Huh7-NTCP cells (*Figure 8A*), we investigated the colocalization between ERp57 and Rab5 (early endosomes), Rab7 (late endosomes), Rab11 (recycling endosomes), or Lamp1 (lysosomes). The quantifications of these results showed that ERp57 could be detected in late endosomes but poorly in the other above-tested locations (*Figure 8B*), in line with the notion that fusion of HBV particles occurs in late endosomes (*Macovei et al., 2013*). Thus, ERp57 can be found at locations compatible for both cell-cell fusion and cell-free entry by internalization.

Altogether, these results indicated that ERp57 is likely a PDI that promotes HBV entry at a membrane fusion step.

Finally, we sought to demonstrate that ERp57 inhibition may prevent HBV propagation in vivo using NTZ, which has a short half-life of about 1.5 hr (*Ruiz-Olmedo et al., 2017*; *Stockis et al., 1996*). We generated a cohort of liver humanized mice (HuHep-mice) derived from the NFRG mouse model (*Azuma et al., 2007*; *Figure 9A*). We retained the animals that displayed >15 mg/ml of human serum albumin (HSA), which corresponds to 40–70% of human hepatocytes in the liver (*Calattini et al., 2015*). In agreement with previous reports (*Perez-Vargas et al., 2019*), these animals supported HBV infection (Group #1) for several weeks/months (*Figure 9B*; see *Figure 9—figure supplement 1* for individual mice). The second group of HuHep mice was treated with NTZ 30 min prior to inoculation with HBV and, then, treated again with NTZ 1 hr later. We found that viremia in this group was delayed by about 4 weeks, as compared to Group #1 for which HBV could disseminate immediately after inoculation. This indicated that the blocking of ERp57 could prevent HBV infection in vivo.

## Discussion

The entry process of enveloped viruses into cells is the series of all events that take place from the attachment of the virus to the host cell until the release of its genome into the cytoplasm. It is a finely regulated and complex process with several steps, in which many viral and cellular factors are involved. The first interaction often occurs with HSPGs. It may lack specificity but serves to give a virus an initial catch hold from which it can recruit specific receptors and entry co-factors that drive the reactions leading to entry. Fusion is the last step of enveloped virus entry and allows the release of the viral capsid into the cytoplasm following the merging of the viral membrane with the membrane of the infected cell. The interactions with the target cell that trigger conformational changes of the viral surface glycoproteins, ultimately leading to the insertion of their fusion peptide into the cell membrane, vary widely for enveloped viruses and can be divided into different scenarios. In the first one (e.g., human immunodeficiency virus), fusion is triggered directly by the interaction of the viral glycoprotein with its cellular receptor, through allosteric conformational rearrangements. In some cases, a sequential interaction with additional host factors is required to trigger the conformational changes required for fusion. In the second scenario (e.g., influenza virus), the interactions with the receptor at the cell surface leads to the endocytosis of viral particles, which is followed by GP

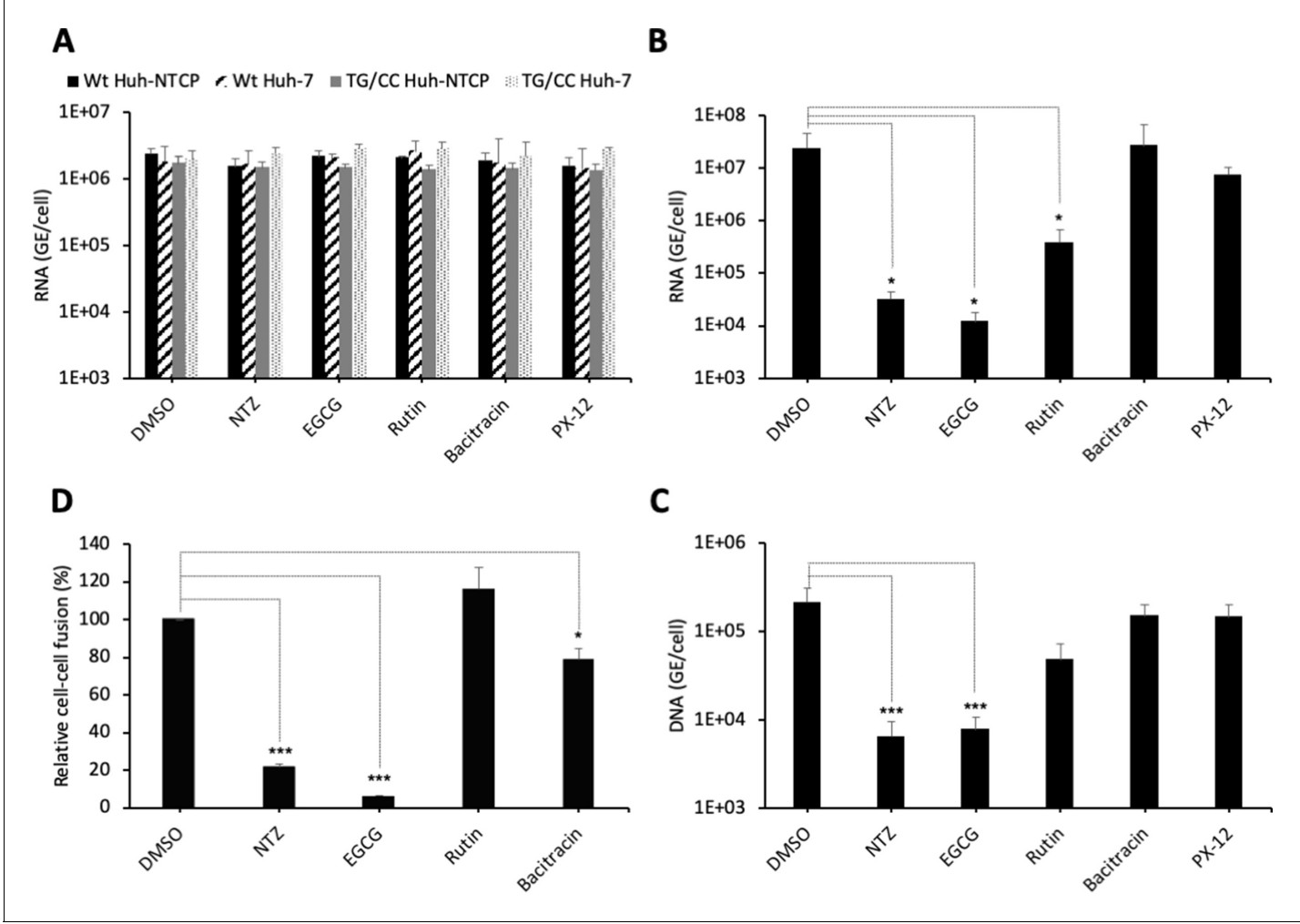

**Figure 6.** PDI inhibitors in HBV entry. (**A**) Hepatitis delta virus (HDV) particles harboring wild-type (wt) or TG/CC mutant (T330C/G308C) hepatitis B virus glycoproteins (HBV GPs) were incubated with Huh7 or Huh7-NTCP cells that were pre-treated for 2 hr with the indicated inhibitors that block different protein disulfide isomerase (PDI) proteins or with dimethyl sulfoxide (DMSO), used as the vehicle. Binding of either virus particles to the cells was quantified by quantitative reverse transcription PCR (RTqPCR) and expressed after normalization with glyceraldehyde 3-phosphate dehydrogenase (GAPDH) RNAs as mean ± SD (N = 3) per ml of cell lysates containing $10^6$ cells. (**B**) HDV or (**C**) HBV particles were used to infect Huh7-NTCP cells that were pre-incubated for 2 hr with the indicated inhibitors that block different PDI proteins or with DMSO, used as a vehicle. Infected cells were grown for 7 days before the total intracellular RNA or DNA was purified. The results of HDV RNA and HBV DNA quantification by RTqPCR and quantitative PCR (qPCR), respectively, are expressed after normalization with GAPDH RNAs as means ± SD (N = 3) per ml of cell lysates containing $10^6$ cells. (**D**) Huh7 'donor' cells co-expressing HBV GPs and a luciferase marker gene driven by the HIV-1 promoter were co-cultured with Huh7-NTCP-tat 'indicator' cells that express HIV Tat protein. The indicated PDI inhibitors were added when 'donor' and 'indicator' cells were mixed for co-cultures and the luciferase activity induced by cell-cell fusion was measured 24 hr later. DMSO was used as a vehicle. Fusion mediated by HBV GPs without inhibitor was taken as 100%. The graphs represent the average of four independent experiments. The PDI inhibitors were used at the following concentrations: nitazoxanide (NTZ), 30 µg/ml; (−)-epigallocatechin 3-gallate (EGCG), 5 µM; rutin, 5 µM; bacitracin, 5 mM; PX-12, 30 µg/ml. No cytotoxicity could be detected in these conditions (*Figure 1—figure supplement 2*).

The online version of this article includes the following source data for figure 6:

**Source data 1.** PDI inhibitors in HBV entry.

protonation in the low-pH environment of the intracellular endosomal organelles that triggers the fusogenic conformational change. In the third scenario (e.g., Ebola virus), the initial interactions of the virion with the cell surface trigger its endocytosis followed by a second interaction with an internal receptor, often found in late endosomes, which is preceded by proteolytic cleavage of the fusion protein by an endosomal protease, leading to fusion activation (*Harrison, 2015*; *White and Whittaker, 2016*). Finally, for certain viruses (e.g., Sindbis virus), the fusion protein requires an activating

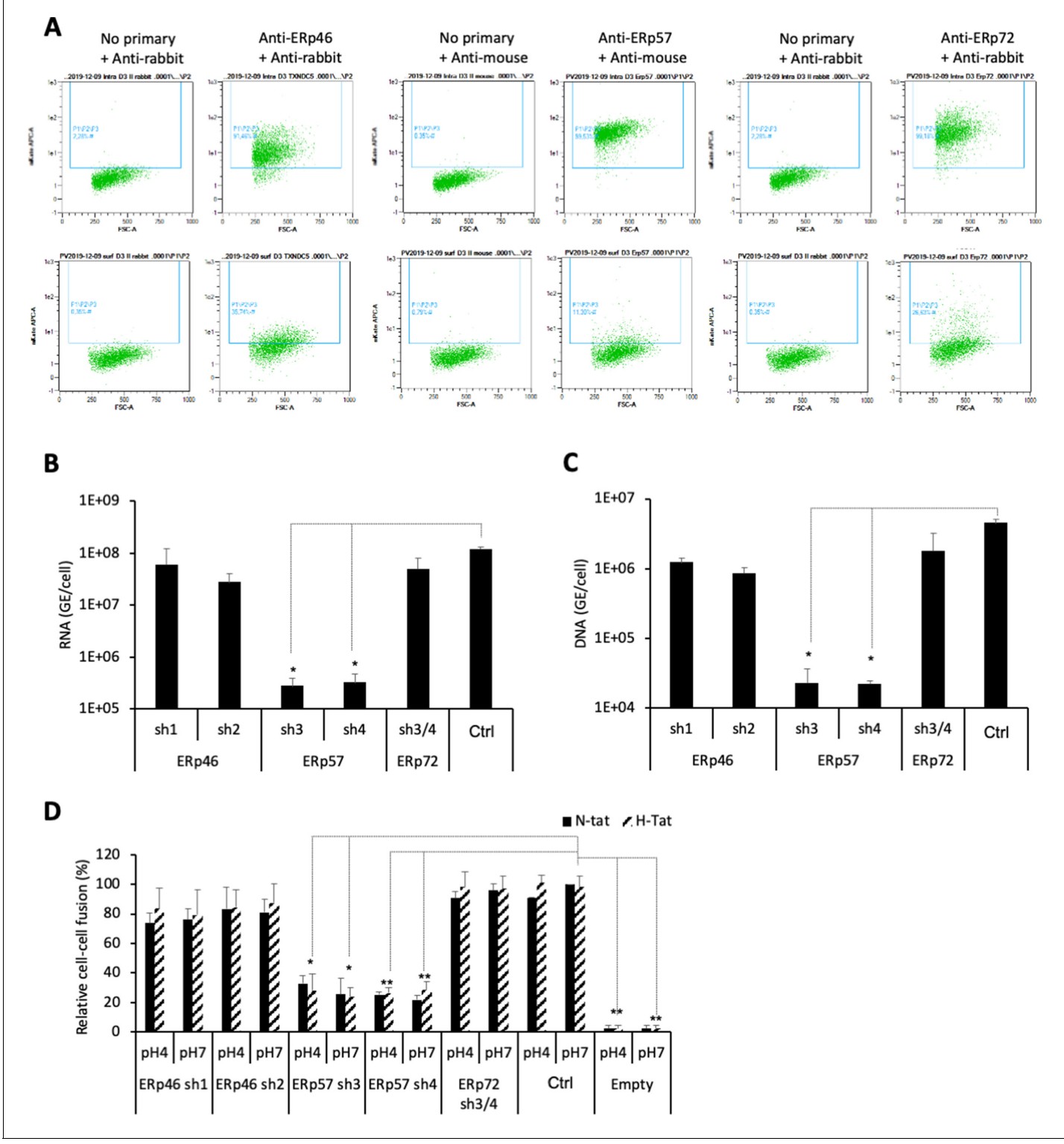

**Figure 7.** ERp57 down-regulation inhibits HBV entry. (**A**) Intracellular (upper panels) and cell-surface (lower panels) staining of ERp46, ERp57, and ERp72 protein disulfide isomerase (PDI) members. Huh7-NTCP cells were subjected to flow cytometry analysis, in order to evaluate the expression of the indicated PDIs. Cells stained with secondary antibody only (no primary) were used to provide the background of flow cytometry analyses. (**B**) Hepatitis delta virus (HDV) or (**C**) hepatitis B virus (HBV) particles were used to infect Huh7-NTCP cells in which the indicated PDIs were down-regulated by lentiviral vectors carrying shRNA (see *Figure 7—figure supplement 1* and *Figure 7—figure supplement 2*). Naive Huh7-NTCP cells were used as controls. Infected cells were grown for 7 days before total intracellular RNA or DNA was purified. The results of HDV RNA and HBV DNA quantification

*Figure 7 continued on next page*

*Figure 7 continued*

by quantitative reverse transcription PCR (RTqPCR) and quantitative PCR (qPCR), respectively, are expressed after normalization with glyceraldehyde 3-phosphate dehydrogenase (GAPDH) RNAs as means ± SD (N = 3) per ml of cell lysates containing $10^6$ cells. (D) Huh7 'donor' cells co-expressing HBV glycoproteins (GPs) and a luciferase marker gene driven by the HIV-1 promoter were co-cultured with Huh7-NTCP-tat 'indicator' cells that express HIV Tat protein in which the indicated PDIs were down-regulated by lentiviral vectors carrying shRNA. After 24 hr, the cells were treated at pH 4 or pH 7 for 3 min. The luciferase activity induced by the fusion between donor and indicator cells was measured 24 hr later. Fusion mediated by HBV GPs at pH 7 with naive Huh7-NTCP-tat cells (Ctrl) was taken as 100%. A control plasmid that does not allow GP expression (Empty) was used to determine the background of luciferase expression. The bars represent the means (N = 3). Error bars correspond to standard deviations.

The online version of this article includes the following source data and figure supplement(s) for figure 7:

**Source data 1.** ERp57 down-regulation inhibits HBV entry.
**Figure supplement 1.** Down-regulation of PDI family members.
**Figure supplement 2.** NTCP expression in target cells.

redox reaction involving disulfide bonds of their glycoproteins to induce membrane fusion (*Key et al., 2015*; *Rey and Lok, 2018*).

Using a cell-cell fusion assay, we found that HBV fusion activity reached similar levels whether indicator cells expressed or not HSPG and/or NTCP but was not increased when the cell co-cultures were exposed at low pH, in contrast to bona fide pH-dependent GPs such as VSV-G or CCHFV Gn/Gc (*Figure 1*). That both HSPG and NTCP, which are respectively HBV virion membrane capture molecules (*Leistner et al., 2008*; *Schulze et al., 2007*) and specific entry factors (*Ni et al., 2014*; *Yan et al., 2015*), are not required for cell-cell fusion highlights that this fusion assay reveals late entry events, such as those occurring after virus interaction with either factor. Similarly, for other viruses such as influenza virus or hepatitis C virus (HCV), binding to their cell entry receptor is not a requirement in both cell-cell fusion (*Lin and Cannon, 2002*) and liposome fusion (*Lavillette et al., 2006*) assays triggered by a low-pH treatment. Thus, while it is clear that cell-cell fusion does not recapitulate per se all the events required to promote cell entry of viral particles since it bypasses the step of internalization that subsequently allows membrane fusion in endosomes that are required for entry of the above-mentioned viruses and of HBV (*Macovei et al., 2013*; *Iwamoto et al., 2019*), it is a suitable experimental tool for investigating some of the structural and functional determinants that promote envelope glycoprotein membrane-fusion activity (*Earp et al., 2004*). Accordingly, our results indicate that the trigger for the HBV membrane fusion mechanism not only is independent of an allosteric interaction of its GPs with the NTCP receptor but also is independent of GP protonation that is induced by the low-pH environment of endosomes. That low pH does not increase HBV cell-cell fusion agrees with previous results indicating that pharmacological agents that raise or neutralize the pH of the endocytic pathway had no effect on HBV infection (*Macovei et al., 2010*; *Macovei et al., 2013*; *Rigg and Schaller, 1992*).

Previous results from *Abou-Jaoudé and Sureau, 2007* showed that cysteine residues of the HBV antigenic loop are essential for HDV infectivity and that viral entry is blocked by inhibitors of thiol/disulfide exchange reaction. Our results extend this notion as they indicate that such reactions seem to be necessary to mediate a critical early post-binding event but not at a later stage of the infection process since no effect in virus infectivity could be detected when DTNB was added at 4 hr post-infection (*Figure 3*). Since isomerization of disulfide bonds has been shown to be crucial for conformational rearrangements of GPs from other enveloped viruses leading to fusion (*Fenouillet et al., 2007*), we sought to investigate if and how such reactions could be implicated during the membrane fusion step of HBV entry. Here, using our cell-cell fusion assay, we found that DTNB blocked HBV GP-mediated membrane fusion (*Figure 3B*). Altogether, these results indicated a role of disulfide bond network of S GP during HBV membrane fusion.

Capitalizing on the above-mentioned experimental information that inhibitors of thiol/disulfide exchange reactions alter virus entry, we sought to examine how disulfide bonds of the HBV GPs, or rather, how a potential reshuffling of its disulfide bond profile, could be important for HBV entry. Indeed, cross-strand disulfides occurring in some viral surface GPs are believed to play a role in virus entry (*Barbouche et al., 2003*; *Jain et al., 2007*; *Rosenthal et al., 1998*; *Wouters et al., 2004*). Particularly, allosteric disulfide bonds can modulate the activity of the proteins in which they reside by mediating a structural change when they are reduced or oxidized (*Hogg, 2003*; *Schmidt et al., 2006*). Allosteric control of protein function is defined as a change in one site (the allosteric site) that

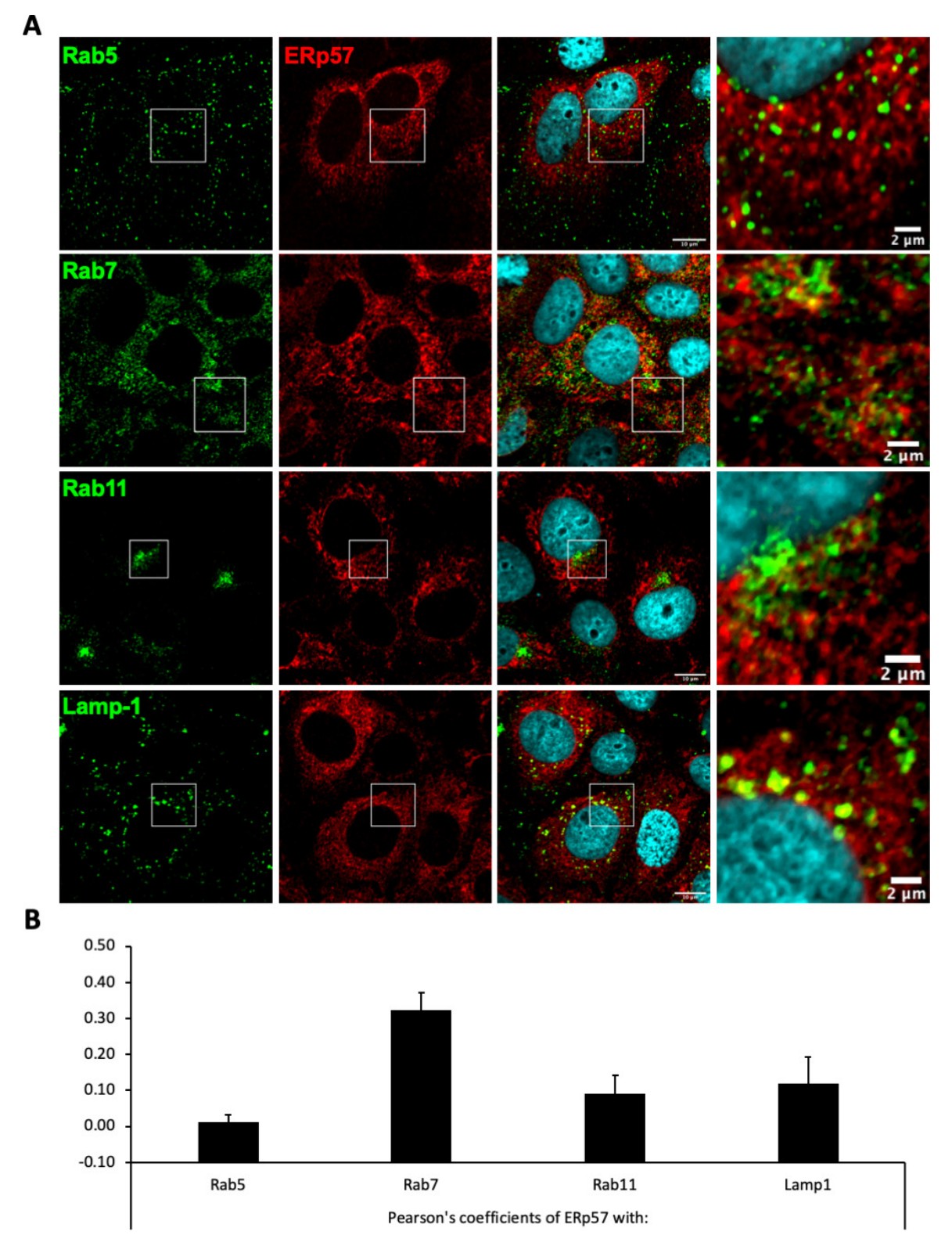

**Figure 8.** Intracellular localization of ERp57 in Huh7-NTCP cells. Huh7-NTCP cells were grown on glass cover slides and fixed 48 hr after seeding. (**A**) Endogenous ERp57 with Rab5, Rab7, Rab11, or Lamp1 were immune-stained, and the colocalization of ERp57 (red channels) with Rab5, Rab7, Rab11, or Lamp1 (green channels) was analyzed by confocal microscopy. Scale bars of panels and zooms from squared area represent 10 μm and 2 μm,

*Figure 8 continued on next page*

*Figure 8 continued*

respectively. (B) The degree of colocalization between ERp57 and the different cell markers was assessed by determining the Pearson's correlation coefficients with the JACoP plugin of ImageJ. Results are expressed as the mean of six individual cells. Error bars correspond to standard deviations.

influences another site by exploiting the protein's flexibility; an allosteric disulfide bond represents

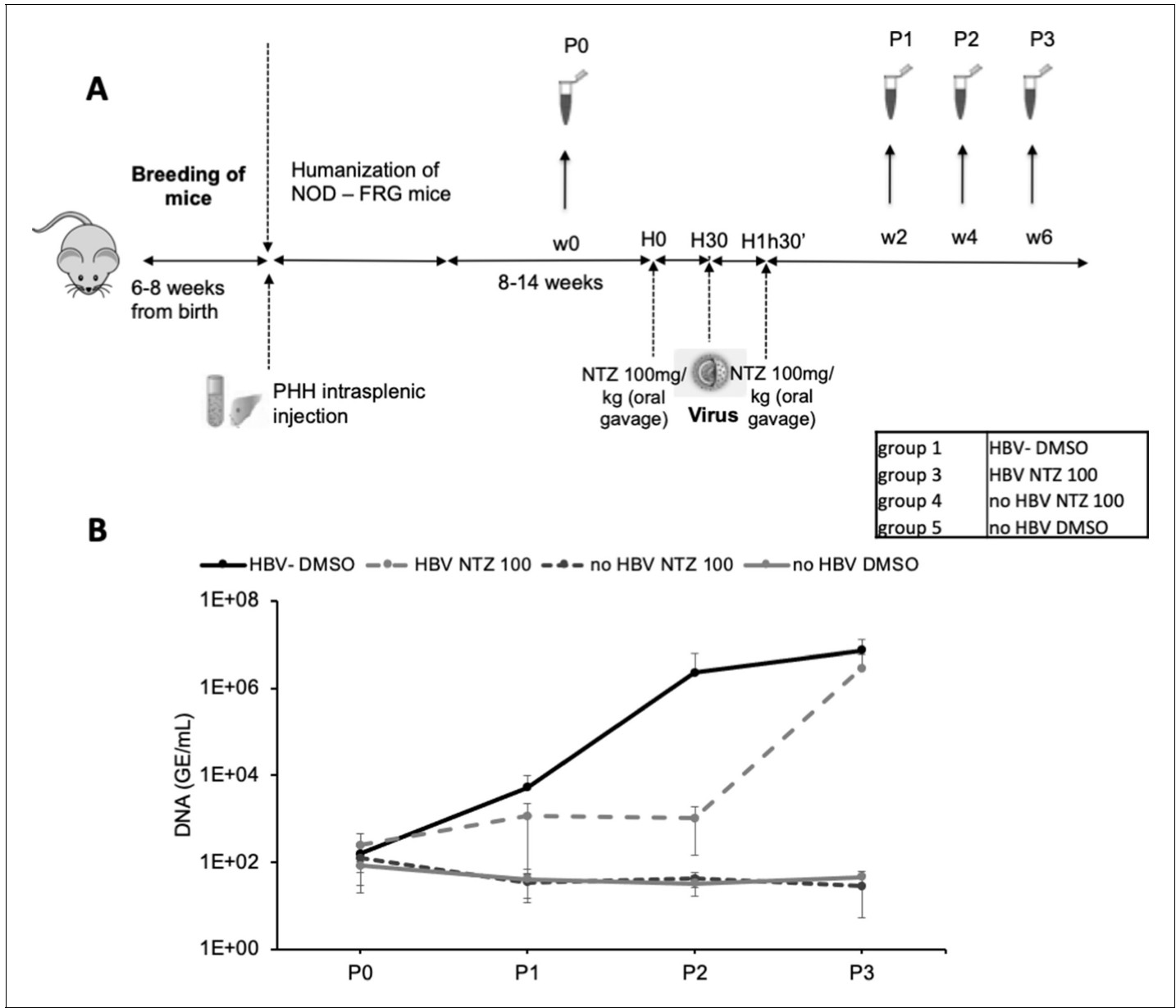

**Figure 9.** In vivo assessment of ERp57 inhibition. (A) 4- to 8-week-old NOD-FRG mice were engrafted with primary human hepatocytes (PHH). After approximately 2–3 months, the animals displaying human serum albumin (HSA) levels >15 mg/ml were randomly split into four different groups (N = 3 to N = 5 animals, see Table in the inset) that were infected with hepatitis B virus (HBV) ($10^8$ genome equivalent (GE)/mouse), using the displayed nitazoxanide (NTZ) treatment schedule. (B) At different time points post-infection, blood samples (50 µl) were collected and the viremia in sera was monitored by quantitative PCR (qPCR) (GE/ml of serum). The graphs show the results of viremia (means ± SD) of HBV. See results of individual mice in *Figure 9—figure supplement 1*.

The online version of this article includes the following source data and figure supplement(s) for figure 9:

**Source data 1.** In vivo assessment of ERp57 inhibition.
**Figure supplement 1.** In vivo assessment of ERp57 inhibition.

the 'allosteric site' and the conformational change triggered by cleavage of such bonds alters protein function. For the HBV S protein, we used the contact prediction method RaptorX (*Ma et al., 2015*; *Wang et al., 2017*) to predict contacts between four Cys-rich regions of the AGL determinant (*Figure 4*), which highlighted that two of these regions may likely interact, that is, the Cys-rich regions III and IV (*Figure 4—figure supplement 1*). Using the secondary structure prediction method JPred (*Cole et al., 2008*), we proposed that these regions organize in two β-strands and we constructed a three-dimensional model of the region 294–317 of the HBV S GP, which indicated that this sequence is compatible with a β-hairpin structural motif containing a CSD bond between C301 and C310 (*Figure 4*). Interestingly, the analysis of the signs of the five χ dihedral angles defined by the Cys residues allowed to classify this particular disulfide bond in a -H Staple conformation, which is a particular type of disulfide geometry associated with allosteric functions that is known to trigger conformational changes upon switching between the reduced and oxidized states (*Chiu and Hogg, 2019*; *Schmidt et al., 2006*). Hence, we hypothesized that the redox state of the C301-C310 disulfide bond may act as an allosteric switch controlling conformational rearrangements of the HBV GP leading, ultimately, to exposure of the fusion peptide. Of note, the β-hairpin region with the predicted CSD lies at the surface of the S protein according to a three-dimensional in silico model (*van Hemert et al., 2008*), which may allow interactions with other HBsAg subunits and/or cellular factors. Yet, to test our structural and dynamic model involving a C301-C310 CSD bond in S GP (*Figure 4*), we reasoned that creating an additional, neighboring disulfide bond between positions 303 and 308 may stabilize the β-hairpin motif (*Figure 4*), which may prevent molecular rearrangements and, thus, allow membrane fusion to occur. The in silico analysis indicates that the T303C/G308C double mutant most probably generates two structural CSD according to our JPred-based (*Cole et al., 2008*) structural modeling, which affects the structural conformation of the C301-C310 CSD that is no longer classified as an allosteric bond. When we tested the T303C/G308C mutation in functional assays, we found that the mutant HBV GPs induced an almost complete loss of infection and fusion activity (*Figure 5*), hence suggesting that by stabilizing cross-strand disulfide exchange, the putative additional disulfide bond prevented conformational rearrangements of HBV GPs that are required for promoting membrane fusion. One possibility is that such stabilization could prevent an isomerization of the C301-C310 CSD bond that generates alternative disulfide bond(s), for example, between C284 and C310, which was proposed in a previous study (*Mangold et al., 1995*). Yet, the antigenic loss of S induced by these mutations did not allow us to design an assay that would detect the additional disulfide bond in the T303C-G308C mutant nor the block of conformational rearrangements that is suggested by its phenotype.

Assuming that the isomerization of the C301-C310 allosteric CSD or of other thiols/disulfides of the AGL determinant could facilitate the conformational rearrangements of HBV GPs required to promote membrane fusion, we hypothesized that such an isomerization could be induced by a host factor from the PDI family, which are enzymes that can both reduce and oxidize disulfide bonds. PDIs consist of a family of 21 structurally related proteins with a thioredoxin-like domain. Most of these isomerases have a CXXC motif that catalyzes formation, reduction, and rearrangement of the disulfide bonds in proteins (*Abell and Brown, 1993*). These isomerases are primarily involved in the folding of proteins in the endoplasmic reticulum (ER), catalyzing formation of their disulfide bonds, and most of these isomerases have ER retention signals. However, some isomerases from the PDI family have also been shown to be present at the cell surface, both in functional and in biochemical assays (*Turano et al., 2002*). Accordingly, cell-surface-localized PDIs are involved in processes such as cell adhesion, nitric oxide signaling, and cell entry of different viruses (*Diwaker et al., 2013*; *Fenouillet et al., 2007*). In support of the notion that PDIs are involved in HBV entry, we found that inhibitors that target different PDI members could block infection and cell-cell fusion though not the binding of viral particles to the cell surface. Of note, we found that bacitracin, which targets PDIA1, did not inhibit HBV entry and membrane fusion, in line with a previous study showing that it could not inhibit HDV entry (*Abou-Jaoudé and Sureau, 2007*). While the above ruled out PDIA1 as an entry co-factor of HBV, we found a strong reduction in the levels of HBV and HDV infection as well as of HBV-induced cell-cell fusion when we used the NTZ and EGCG inhibitors (*Figure 6*), which target ERp57 (*Müller et al., 2008*; *Pacello et al., 2016*). Consistently, we detected a low but significant expression of ERp57 as well as of ERp46 and ERp72 at the cell surface (*Figure 7*), in agreement with a previous study (*Turano et al., 2002*). Furthermore, we detected ERp57 in late endosomes (*Figure 8*), which is meaningful since previous reports have shown that HBV infection of HepaRG cells

depends on Rab5 and Rab7 (*Macovei et al., 2013*), which are GTPases involved in the biogenesis of endosomes, and that the epidermal growth factor receptor is a host-entry cofactor that interacts with NTCP and mediates HBV internalization (*Iwamoto et al., 2019*). Using a gene silencing approach, we confirmed that down-regulation of ERp57 but not of these alternative PDIs could decrease the levels of HDV and HBV infection as well as of cell-cell fusion (*Figure 7*). Importantly, we showed that a short-time treatment of liver humanized mice with NTZ could delay HBV infection by approximately 2–4 weeks (*Figure 9*). Since NTZ has a short half-life of about 1.5 hr in vivo (*Ruiz-Olmedo et al., 2017*; *Stockis et al., 1996*) and since NTZ was administrated at very short times before and after HBV inoculation, we calculated that less than 10% of the drug was still present in those mice at 7 hr post-infection, which likely precludes an effect on HBV post-entry steps (*Korba et al., 2008*). Altogether, these results support the role of ERp57 at early steps of HBV infection and validate this PDI as a therapeutic target. Note that our results did not discard the possibility that some other PDIs could also play a role during HBV entry into cells.

The fusion-mediating GPs of enveloped viruses contain a sequence, termed fusion peptide, that interacts with and destabilizes the cellular target membrane. Such an event is finely controlled so as to occur at the appropriate time and location and to prevent fortuitous inactivation of GP fusion activity and virus infectivity. Hence, a conformational change in these GPs is a requirement to induce the accessibility and function of the fusion peptide segment. Candidate fusion peptides are generally identified as hydrophobic sequences, of approximately 16–26 residues in length, that are conserved within a virus family and that may adopt α-helical conformation with strongly hydrophobic faces. They can be internal or located at the amino-terminus of fusion GP subunits (*Apellániz et al., 2014*; *Epand, 2003*; *Martin et al., 1999*). There are a number of criteria that characterize fusion peptide segments and, while none of these criteria taken individually are absolute to define a fusion peptide segment, they are sufficiently restrictive to predict if a given region of a protein presents features of a fusion peptide segment (*Delos and White, 2000*; *Delos et al., 2000*), which needs to be further functionally tested.

Previously, a conserved peptide comprising 23 amino acids at the N-terminal end of the HBV S protein and overlapping its TM1 sequence was shown to interact with model membranes, causing liposome destabilization in a pH-dependent manner (*Berting et al., 2000*; *Rodríguez-Crespo et al., 1994*; *Rodríguez-Crespo et al., 1995*; *Rodríguez-Crespo et al., 1999*). However, it was also demonstrated that hydrophobic residues in TM1 were critical for S protein expression as well as for infectivity (*Siegler and Bruss, 2013*). An essential role of TM1 in fusion mechanism, albeit in a pH-independent manner, could be shown for the duck hepatitis B virus (DHBV) (*Chojnacki et al., 2005*; *Grgacic and Schaller, 2000*), although there is also evidence for the involvement of the preS domain of DHBV at an early step of infection, likely during the fusion process (*Delgado et al., 2012*).

Here, through a computational hydropathy analysis of the HBV GPs, we identified two potential short sequences within the preS1 and preS2 regions that may potentially interact with membrane bilayers. To validate these predictions, we characterized in both infection and cell-cell fusion assays HBV GP mutants in key positions in either sequence. We found that while none of the mutations in the preS2 segment altered infection or membrane fusion activities, mutations in the preS1 sequence induced an almost complete loss of infectivity and cell-cell fusion (*Figure 2*). Note that these mutants had similar if not identical levels of cell-surface-expressed L, M, and S proteins and/or capacity to induce the formation of HDV particles. These results suggested that the preS1 region harbors a fusion peptide in addition to the NTCP-binding determinant.

Overall, our study characterizes some crucial determinants of HBV entry and membrane fusion. The mechanism by which fusion proteins are activated and undergo conformational rearrangements or fusion intermediates is a particularly complex process involving several regions of viral surface GPs. Our results suggest that for HBV, this mechanism could be triggered by ERp57, allowing a thiol/disulfide exchange reaction to occur and regulate isomerization of critical CSD(s), which ultimately results in the exposition of the fusion peptide that seems to be located within the preS1 region.

## Materials and methods

### Key resources table

| Reagent type (species) or resource | Designation | Source or reference | Identifiers | Additional information |
|---|---|---|---|---|
| Strain, strain background (*Mus musculus*, females and males) | NOD-FRG mice | DOI: 10.1038/nbt1326 DOI : 10.1074/jbc. M115.662999 | | Breeding and experimentation in PBES – originally purchased to YEcuris corporation |
| Strain, strain background (HBV) | Hepatitis B virus (HBV) | This paper | | HBV, genotype D, produced by co-transfection of HepG2.2.15 cells with plasmids pCiHB(env-) and pT7HB2.7 |
| Strain, strain background (HDV) | Hepatitis D virus (HDV) | This paper | | HDV, genotype 1, produced by co-transfection of Huh7 cells with plasmids pSVLD3 and pT7HB2.7 or variant constructs |
| Cell line (*Homo sapiens*) | Huh7 - hepatocarcinoma cells | PMID:6286115 | | |
| Cell line (*Homo sapiens*) | Huh7-NTCP | This paper | | Generated by transduction with pLX304NTCP retroviral vector and selection with blasticidin |
| Cell line (*Homo sapiens*) | Huh7-Tat (H-tat) cells | This paper | | Generated by transduction with LXSN-tat retroviral vector and selection with G418 |
| Cell line (*Homo sapiens*) | H-tat cells down-regulated for ERp46, ERp57, or ERp72 | This paper | | Generated by transduction of H-tat cells with shRNA lentiviral vectors against ERp46, ERp57, or ERp72 followed by selection with puromycin |
| Cell line (*Homo sapiens*) | Huh7-NTCP-Tat (N-tat) cells | This paper | | Generated by transduction of Huh7-NTCP cells with LXSN-tat retroviral vector |
| Cell line (*Homo sapiens*) | N-tat cells down-regulated for ERp46, ERp57, or ERp72 | This paper | | Generated by transduction of N-tat cells with shRNA lentiviral vectors against ERp46, ERp57, or ERp72 followed by selection with puromycin |
| Cell line (*Homo sapiens*) | HepG2.2.15 human hepatoma cells | From David Durantel lab | | Production of HBV particles |
| Cell line (*Homo sapiens*) | 293T human kidney cells | ATCC | CRL-1573 | Production of retro- and lentiviral particles |
| Cell line (*Cricetulus griseus*, female) | CHO-K1 Chinese hamster ovary cells | ATCC | CCL-61 | Cell-cell fusion assays |
| Transfected construct (human) | pLX304NTCP | DNASU plasmid repository | HQ447437 | Retroviral construct to transfect and express NTCP |
| Transfected construct (HBV) | pSVLD3 | DOI: 10.1128/JVI.63. 5.1945–1950.1989 | | Harbors a trimer of the HDV, genotype 1 genome. Used for production of HDV particles |

*Continued on next page*

*Continued*

| Reagent type (species) or resource | Designation | Source or reference | Identifiers | Additional information |
|---|---|---|---|---|
| Transfected construct (HBV) | pT7HB2.7 | DOI: 10.1128/JVI.68.6.4063–4066.1994 | | Gift from Camille Sureau, used for production of HBV and HDV particles and expression of HBV envelope proteins |
| Transfected construct (HBV) | pT7HB2.7Mless (noM) | This paper | | Generated for expression of HBV L and S proteins (M protein is silenced) |
| Transfected construct (HBV) | pCiL | DOI: 10.1128/JVI.77.9.5519–5523.2003 | | Encodes only the L-HBsAg protein |
| Transfected construct (HBV) | pCIS | DOI: 10.1128/JVI.80.10.4648–4655.2006 | | Encodes only the S-HBsAg protein |
| Transfected construct (CCHFV) | pCAGGS_GP/wt-M | DOI: 10.1128/JVI.03691–14 | | Major open reading frame of CCHFV M-segment subcloned into pCAGGS |
| Transfected construct (HBV) | pCIHB(env-) | DOI: 10.1128/JVI.00621–06 | | Gift from Camille Sureau, used for production of HBV particles |
| Transfected construct (HIV1-Tat) | LXSN-tat retroviral vector | DOI: 10.1128/JVI.73.3.1956–1963.1999 | | HIV-1 *tat* gene cloned into the LXSN retroviral vector |
| Transfected construct (HIV1-LTR) | pLTR-luc | DOI: 10.1016/0378-1119(90)90032 m | | Gift from Olivier Schwartz, contains a 722-base pair *Xho*I (−644)-*Hind*III (+78) fragment from HIV-1 placed in front of the luciferase reporter gene |
| Transfected construct (VSV) | phCMV-VSV-G | DOI: 10.1016/s0091-679x(08)60600–7 | | To express the envelope protein of VSV |
| Transfected construct (human) | shRNA against ERp46 (ERp46-shRNA 1) | Sigma | NM_022085 / TRCN0000064353 / PLKO.1 | Lentiviral construct to transfect and express the shRNA |
| Transfected construct (human) | shRNA against ERp46 (ERp46-shRNA 2) | Sigma | NM_022085 / TRCN0000064354 / PLKO.1 | Lentiviral construct to transfect and express the shRNA |
| Transfected construct (human) | shRNA against ERp57 (ERp57-shRNA 3) | Sigma | NM_005313 / TRCN0000319038 / PLKO | Lentiviral construct to transfect and express the shRNA |
| Transfected construct (human) | shRNA against ERp57 (ERp57-shRNA 4) | Sigma | NM_005313 / TRCN0000147738 / PLKO.1 | Lentiviral construct to transfect and express the shRNA |
| Transfected construct (human) | shRNA against ERp72 (ERp72-shRNA 3) | Sigma | NM_004911 / TRCN0000289676 / PLKO.1 | Lentiviral construct to transfect and express the shRNA |
| Transfected construct (human) | shRNA against ERp72 (ERp72-shRNA 4) | Sigma | NM_004911 / TRCN0000049334 / PLKO.1 | Lentiviral construct to transfect and express the shRNA |
| Transfected construct (human) | shRNA against ERp72 (ERp72-shRNA 5) | Sigma | NM_004911 / TRCN0000307107 / PLKO.1 | Lentiviral construct to transfect and express the shRNA |
| Biological sample (*M. musculus*) | Blood samples | PBES (Plateau de Biologie Experimentale de la Souris) SFR Biosciences Lyon | | Isolated from NOD-FRG mice |
| Antibody | Anti-HBsAg antibody, HPR conjugated (goat polyclonal) | DiaSorin | 9F80-01 | WB (1:400) |

*Continued on next page*

*Continued*

| Reagent type (species) or resource | Designation | Source or reference | Identifiers | Additional information |
|---|---|---|---|---|
| Antibody | Anti-human calnexin (rabbit polyclonal) | Enzo | ADI-SPA-865-F | WB (1:1000) |
| Antibody | Anti-mouse TXNDC5/ERp46 (rabbit polyclonal) | Abcam | Ab10292 | FACS (1:20) WB (1:1000) |
| Antibody | Anti-human ERp57 (mouse monoclonal) | Abcam | Ab13506 | FACS (2 μg/10$^6$ cells) WB (1:10,000) IF (1:100) |
| Antibody | Anti-human ERp72 (rabbit polyclonal) | Abcam | Ab155800 | FACS (1:100) WB (1:1000) |
| Antibody | Anti-human NTCP/SLC10A1 antibody, PE conjugated (rabbit polyclonal) | Bioss Antibodies | bs-1958R-PE | FACS (1:100) |
| Antibody | Anti-human Rab5 (rabbit monoclonal) | Cell Signaling Technology | (C8B1):3547 | IF (1:200) |
| Antibody | Anti-human Rab7 (rabbit monoclonal) | Cell Signaling Technology | (D95F2):9367 | IF (1:100) |
| Antibody | Anti-human Rab11 (rabbit monoclonal) | Cell Signaling Technology | (D4F5):5589 | IF (1:50) |
| Antibody | Anti-human Lamp1 (rabbit monoclonal) | Cell Signaling Technology | (D2D11):9091 | IF (1:200) |
| Sequence-based reagent | F52A | This paper | preS1 mutagenesis PCR primers | GTAGGAGCTGGAGCAG CCGGGCTGGGTTTCAC |
| Sequence-based reagent | F52E | This paper | preS1 mutagenesis PCR primers | GTAGGAGCTGGAGCAGA AGGGCTGGGTTTCAC |
| Sequence-based reagent | G53A | This paper | preS1 mutagenesis PCR primers | CTGGAGCATTCGCGCT GGGTTTCAC |
| Sequence-based reagent | F56A | This paper | preS1 mutagenesis PCR primers | TTCGGGCTGGGTGCC ACCCCACCGCA |
| Sequence-based reagent | W66A | This paper | preS1 mutagenesis PCR primers | GAGGCCTTTTGGGGGCG AGCCCTCAGGCTC |
| Sequence-based reagent | W66E | This paper | preS1 mutagenesis PCR primers | GAGGCCTTTTGGGGGAG AGCCCTCAGGCTC |
| Sequence-based reagent | Y129A | This paper | preS2 mutagenesis primers | GAGTGAGAGGCCTGGCTT TCCCTGCTGGTG |
| Sequence-based reagent | F130A | This paper | preS2 mutagenesis primers | GAGAGGCCTGTATGCCCC TGCTGGTGG |
| Sequence-based reagent | S136E | This paper | preS2 mutagenesis primers | CCCTGCTGGTGGCTCCGAA TCAGGAACAGTAAAC |
| Sequence-based reagent | L144A | This paper | preS2 mutagenesis primers | CAGTAAACCCTGTTGCGACT ACTGCCTCTCC |
| Sequence-based reagent | T303C | This paper | CSD mutagenesis primers | CCTCCTGTTGCTGTTGCAAA CCTTCGGACG |
| Sequence-based reagent | G308C | This paper | CSD mutagenesis primers | GTACCAAACCTTCGGACTGT AATTGCACCTGTATTCCC |
| Sequence-based reagent | TG/CC | This paper | CSD mutagenesis primers | GTTGCAAACCTTCGGACTGT AATTGCACCTGTATTCCC |
| Commercial assay or kit | FuGENE HD Trasnfection Reagent | Promega | E2312 | Transfection reagent |

*Continued on next page*

*Continued*

| Reagent type (species) or resource | Designation | Source or reference | Identifiers | Additional information |
|---|---|---|---|---|
| Commercial assay or kit | Dual-Luciferase Reporter Assay System | Promega | E1910 | Quantification of luciferase activity |
| Commercial assay or kit | iScript cDNA synthesis kit | Bio-Rad | 1708891 | cDNA synthesis |
| Commercial assay or kit | FastStart Universal SYBR Green Master | Roche Sigma | 4913850001 | Real-time qPCR assays |
| Commercial assay or kit | CytoTox-ONE Homogen Membrane Integrity Assay | Promega | G7891 | Cytotoxicity assay |
| Chemical compound, drug | Bacitracin | Sigma | B0125-250KU | Water |
| Chemical compound, drug | NTZ (nitazoxanide) | Sigma | N0290-50MG | DMSO |
| Chemical compound, drug | EGCG ((−)-epigallocatechin gallate) | Sigma | E4268-100MG | Water |
| Chemical compound, drug | Rutin Hydrate | Sigma | R5143-50G | DMSO |
| Chemical compound, drug | PX-12 | Sigma | M5324-5MG | DMSO |
| Chemical compound, drug | DTNB (5,5′-dithiobis(2-nitrobenzoic acid)) | Sigma | D218200-1G | DMSO |
| Chemical compound, drug | EZ-Link Sulfo-NHS-LC-LC-Biotin | Life technologies | 21338 | |
| Software, algorithm | ImaJ software | ImaJ | RRID:SCR_003070 | |
| Software, algorithm | Membrane Protein eXplorer | http://blanco.biomol.uci.edu/mpex/ | RRID:SCR_014077 | |
| Software, algorithm | RaptorX | http://raptorx.uchicago.edu/ | RRID:SCR_018118 | |
| Software, algorithm | Jpred | http://www.compbio.dundee.ac.uk/jpred/ | RRID:SCR_016504 | |
| Software, algorithm | MODELLER | http://salilab.org/modeller/modeller.html | RRID:SCR_008395 | |
| Software, algorithm | Clustal X | http://www.clustal.org/clustal2/ | RRID:SCR_017055 | |
| Software, algorithm | Molecular Modelling Toolkit | http://dirac.cnrs-orleans.fr/MMTK.html | | |
| Software, algorithm | GROMACS | http://www.gromacs.org | RRID:SCR_014565 | |
| Software, algorithm | UCSF Chimera | http://plato.cgl.ucsf.edu/chimera/ | RRID:SCR_004097 | |
| Other | Hoechst 33342 stain | Thermo Fisher | H3570 | 10 µg/ml |

*Continued on next page*

*Continued*

| Reagent type (species) or resource | Designation | Source or reference | Identifiers | Additional information |
|---|---|---|---|---|
| Other | Streptavidin Agarose Resin | Thermo Fisher | 20353 | |
| Other | TRI-Reagent | Molecular Research Center Euromedex | TR118-200 | RNA extraction |

## Plasmids

Plasmid pSVLD3 harboring a trimer of the HDV gt1 genome (accession number M21012.1), pCiL encoding the L protein, pCiS encoding the S protein (*Komla-Soukha and Sureau, 2006*), and pT7HB2.7 encoding the three HBV envelope proteins were a gift from *Sureau, 2010*; *Sureau et al., 1994*. To induce the expression of L and S only, the pT7HB2.7 plasmid was modified at the M start codon and Kozak consensus sequence in order to silence the expression of M protein, resulting in pT7HB2.7Mless construct. The pCiM plasmid encoding the M protein was constructed by deleting the preS1 region from pCiL until the N-terminal methionine of preS2. All mutations in pT7HB2.7 plasmid were introduced by point-directed mutagenesis. The phCMV-VSV-G encoding the G protein from VSV and pCAGGS-GP/wt-M encoding the Gn and Gc glycoproteins from CCHFV were described previously (*Freitas et al., 2020*). The plasmid encoding the luciferase reporter under control of an HIV-1 long terminal repeat internal promoter (pLTR-luc) was a gift from Olivier Schwartz and was used as described before (*Lavillette et al., 2007*). Mission shRNA plasmids (Sigma), shRNA sequences, and oligonucleotides used for introducing mutations in HBV GPs are described in *Supplementary file 1*.

## Cells

Huh7 human hepatocarcinoma cells and Huh7-NTCP cells, which were generated by transduction of Huh7 cells with a retroviral vector transducing the NTCP plasmid (pLX304NTCP, DNASU) and selected for blasticidin resistance, were grown in William's E medium (WME) (Gibco, France) supplemented with non-essential amino acids, 2 mM L-glutamine, 10 mM 4-(2-hydroxyethyl)-1-piperazineethanesulfonic acid (HEPES) buffer, 100 U/ml of penicillin, 100 µg/ml of streptomycin, and 10% fetal bovine serum. 293T human kidney cells (ATCC CRL-1573), CHO-K1 (CHO) Chinese hamster ovary cells (ATCC CCL-61), and CHO-pgsB-618 cells (ATCC CRL-2241), which do not produce glycosaminoglycans, were grown in Dulbecco's modified minimal essential medium (DMEM, Gibco) supplemented with 100 U/ml of penicillin, 100 µg/ml of streptomycin, and 10% fetal calf serum. Huh7-tat and Huh7-NTCP-tat indicator cells expressing HIV Tat were generated by transduction of Huh7 and Huh7-NTCP cells, respectively, with the LXSN-tat retroviral vector and selected for G418 resistance. HepG2.2.15 human hepatoma cells were used to produce HBV virus and were maintained in WME medium complemented with 10% fetal bovine serum. Authentication of purchased cell lines was performed by ATCC. Authentication of Huh7 cells was based on expression of human transferrin and serum albumin. Authentication of HepG2.2.15 cells was based on titration of released infectious HBV particles. All cell lines were certified mycoplasma-free, as per our monthly contamination testing.

## PDI inhibitors

DTNB, NTZ, EGCG, rutin, bacitracin, and PX-12 were purchased from Sigma-Aldrich and dissolved in dimethyl sulfoxide (DMSO), ethanol, or water according to the manufacturer's instructions.

## Antibodies

For western blot analysis, HBs antigen and calnexin were detected with goat anti-HBV polyclonal antibody (Murex, DiaSorin) coupled to horseradish peroxidase (HRP) and rabbit calnexin polyclonal antibody (Enzo), respectively. The rabbit anti-ERp46 (Abcam), mouse anti-ERp57 (Abcam), and mouse anti-ERp72 (Santa Cruz Biotechnology) antibodies were used for detecting PDI proteins by

flow cytometry and western blot. NTCP was detected with polyclonal NTCP/SLC10A1 antibody (Bioss Antibodies) coupled to Phycoerythrin (PE) for flow cytometry. Mouse anti-ERp57 (Abcam), Rabbit anti-Rab5, anti-Rab7, anti-Rab11, and anti-Lamp1 (Cell Signaling Technology), and Donkey anti-Rabbit-Alexa-Fluor-488 and Donkey anti-Mouse-Alexa-Fluor-568 (Thermo Fisher) antibodies were used for immunofluorescence (IF) studies.

### shRNA-expressing stable cell lines

293 T cells were seeded 24 hr prior to transfection with VSV-G plasmid, psPAX2 packaging plasmid, and pLKO.1 expression vector carrying shRNA against ERp46, ERp57, or ERp72 using calcium phosphate precipitation. Medium was replaced 16 hr post-transfection. Vector supernatants were harvested 24 hr later, filtered through a 0.45 µm filter. Stable knockdown of ERp72, ERp57, or ERp46 in Huh7-NTCP, Huh7-tat, and Huh7-NTCP-tat cells was performed by selection with puromycin after lentiviral transduction. The knockdown was validated by flow cytometry and western blot using antibodies against ERp46, ERp57, or ERp72.

### Cell-cell fusion assays

Huh7 'donor' cells ($2.5x10^5$ cells/well seeded in six-well tissue culture dishes 24 hr prior to transfection) were co-transfected using FuGENE six transfection reagent (Promega) with 3 µg of pT7HB2.7-wt or mutated glycoproteins and 50 ng of pLTR-luc reporter plasmid. For a positive control, cells were co-transfected with 3 µg of either pCAGGS-GP/wt-M, expressing CCHFV GPs, or 1 µg of phCMV-VSV-G and with 50 ng of the pLTR-luc plasmid. For negative controls, cells were co-transfected with 2 µg of an empty phCMV plasmid and 50 ng of the pLTR-luc plasmid. 12 hr later, transfected cells were detached with Versene (0.53 mM ethylenediaminetetraacetic acid (EDTA); Gibco), counted, and reseeded at the same concentration ($10^5$ cells/well) in 12-well plates. Huh7-tat or Huh7-NTCP-tat indicator cells, detached with EDTA and washed, were then added to the transfected cells ($3x10^5$ cells per well). After 24 hr of cocultivation, the cells were washed with phosphate-buffered saline (PBS), incubated for 3 min in fusion buffer (130 mM NaCl, 15 mM sodium citrate, 10 mM MES [2-(N-morpholino)ethanesulfonic acid], 5 mM HEPES) at pH 4, pH 5, or pH 7, and then washed three times with normal medium. The luciferase activity was measured 24 hr later using a luciferase assay kit according to the manufacturer's instructions (Promega).

### HDV particle production and infection

Huh7 cells were seeded in 10 cm plates at a density of $10^6$ cells per plate and were transfected with a mixture of 2.5 µg of pSVLD3 plasmid and 10 µg of plasmid allowing the expression of surface envelope glycoproteins of VSV or HBV using FuGENE six transfection reagent (Promega), as described previously (*Perez-Vargas et al., 2019*). Transfected cells were grown for up to 9 days in primary hepatocyte maintenance medium containing 2% DMSO to slow cell growth.

The supernatants of virus producer cells were filtrated through 0.45-nm-pore filters and were analyzed by quantitative reverse transcription PCR (RTqPCR) for detection of HDV RNA, using the primers described below. These supernatants were also used for infection experiments in Huh7-NTCP cells or PDI-down-regulated Huh7-NTCP cells, which were seeded in 48-well plates at a density of $1.5x10^4$ cells per well. Infected cells were cultured in primary hepatocyte maintenance medium containing 2% DMSO following infection. RTqPCR assays were used to assess infectivity of viral particles at 7 days post-infection.

For inhibition assays, drugs were incubated with cells for 2 hr at 37°C before virus addition or at different times post-infection and the infectivity was assessed 7 days post-infection by RT-qPCR.

### Binding assays

HDV wt particles ($10^7$ genome equivalent [GE]) were added to Huh7-NTCP cells and incubated for 1 hr at 4°C. Unbound virions were removed by three washes with cold PBS, and RTqPCR was used to assess the amount of bound viral particles.

### RTqPCR detection of HDV RNAs in virus producer and infected cells

Cells were washed with PBS and total RNA was extracted with TRI Reagent according to the manufacturer's instructions (Molecular Research Center). RNAs were reverse transcribed using random

oligonucleotide primers with iScript (Bio-Rad). The following specific primers were used: for HDV RNA quantification, forward primer 5'-GGACCCCTTCAGCGAACA and reverse primer 5'-CCTAGCA TCTCCTCCTATCGCTAT. Quantitative PCR (qPCR) was performed using FastStart Universal SYBR Green Master (Roche) on a StepOne Real-Time PCR System (Applied Biosystems). As an internal control of extraction, in vitro-transcribed exogenous RNAs from the linearized Triplescript plasmid pTRI-Xef (Invitrogen) were added to the samples prior to RNA extraction and quantified with specific primers (5'-CGACGTTGTCACCGGGCACG and 5'-ACCAGGCATGGTGGTTACCTTTGC). All values of intracellular HDV RNAs were normalized to glyceraldehyde 3-phosphate dehydrogenase (GAPDH) gene transcription. For GAPDH mRNA quantification, we used the forward 5'-AGGTGAAGG TCGGAGTCAACG and reverse 5'-TGGAAGATGGTGATGGGATTTC primers.

## Western blot analyses

The proteins from pelleted cell supernatants or extracted from total cell lysates were denatured in Lammeli buffer at 95°C for 5 min and were separated by sodium dodecyl sulfate polyacrylamide gel electrophoresis (SDS–PAGE) and then transferred to nitrocellulose membranes (GE Healthcare). Membranes were blocked with 5% nonfat dried milk in PBS and incubated at 4°C with a rabbit or mouse antibody diluted in PBS-0.01% milk, followed by incubation with an IRdye secondary antibody (Li-Cor Biosciences). Membrane visualization was performed using an Odyssey infrared imaging system CLx (LI-COR Biosciences).

For cell-surface biotinylation, Huh7 cells were transfected into 10 cm plates with plasmids encoding wt or mutant HBV GPs. After 48 hr, the cell monolayers were rinsed three times with ice-cold PBS and overlaid with 0.5 ml biotin solution (0.5 mg sulpho-N-hydroxysuccinimide–biotin (Pierce) per ml of PBS, pH 7.2). The cells were then labeled for 30 min at 4°C. The biotin solution was removed and the cells were rinsed once with ice-cold 100 mM glycine solution and then incubated for 15 min with 100 mM glycine at 4°C to stop the reaction. The last washing step was performed with ice-cold PBS. Proteins were solubilized by the addition of 1 ml radioimmunoprecipitation assay buffer and equivalent quantities of protein lysates from each sample (Nanodrop quantification; Thermofisher) were immunoprecipitated with biotin-agarose beads. Proteins were electrophoresed under reducing conditions in SDS–PAGE followed by electrophoretic transfer to nitrocellulose. Surface-biotinylated proteins were detected with anti-HBV antibody (Murex) coupled to HRP and enhanced chemiluminescence (ECL; Roche). The membranes of biotinylated samples were routinely re-probed with anti-calnexin antibody to confirm the absence of the intracellular protein calnexin. In addition, 10% of each lysate was denatured and loaded onto separate gels. Immunoblotting for calnexin on the membranes of lysate was done to confirm uniform protein loading.

Densitometry analysis (Image Lab BioRad software) was used to estimate the relative total amount of L, M, and S mutant proteins, which were expressed relative of the wild-type L, M, and S total proteins.

## Flow cytometry

The surface expression of NTCP, ERp46, ERp57, and ERp72 was quantified by fluorescence activated cell sorting (FACS) analysis from $10^6$ live cells using antibodies added to cells for 1 hr at 4°C. After washing, the binding of antibody to the cell surface was detected using PE (phycoerythrin)-conjugated anti-mouse antibodies.

## Immunofluorescence, confocal microscopy imaging, and image analysis

Huh7-NTCP cells were grown on uncoated 14-mm-diameter glass coverslips. 48 hr after seeding, cells were washed with PBS, fixed with 3% paraformaldehyde in PBS for 15 min, quenched with 50 mM NH$_4$Cl, and permeabilized with 0.1% Triton X-100 for 7 min. Fixed cells were then incubated for 1 hr with primary antibodies in 1% bovine serum albumin (BSA)/PBS, and washed and stained for 1 hr with the corresponding fluorescent Alexa Fluor conjugated secondary antibody (Alexa-Fluor-488 and Alexa-Fluor-568; Thermo Fisher) in 1% BSA/PBS. Cells were washed three times with PBS, stained for nuclei with Hoechst 33342 (Thermo Fisher) for 5 min, washed two times with PBS, and mounted on Mowiol 40–88 (Sigma-Aldrich) prior to image acquisition with LSM-710 confocal microscope (Zeiss). Single-section confocal images of 0.6 μm thickness were analyzed with the ImageJ

software. The Pearson's correlation coefficients were calculated by using the JACoP plugin for ImageJ.

## Cytotoxicity assay

The release of lactate dehydrogenase (LDH) from damaged cells was measured with CytoTox-ONE (Promega, MA, USA) homogeneous membrane integrity assay. Cells were grown in a 96-well flat-bottom culture plate at a density of $3x10^3$ cells per well and treated with the different drugs for 2 hr or 24 hr. Maximum LDH release was determined by adding 2 µl of CytoTox-ONE lysis solution to control wells for 10 min. The assay was performed in 96-well plates by adding 100 µl of the sample supernatant and 100 µl of CytoTox-ONE reagent, after which the plate was shaken for 10 s. After 10 min of incubation, 50 µl CytoTox-One stop solution was added and the plate was shaken again for 10 s. The fluorescence signal was measured at $\lambda_{EX}$ = 560 nm, $\lambda_{EM}$ = 590. LDH release was calculated as the percentage of LDH released in the culture media of total LDH (media and lysates).

## Fusion peptide prediction

The HBV surface sequence used was taken from the UniProt database, with accession number P03138. Hydropathy plots were obtained with Membrane Protein eXplorer software (*Snider et al., 2009*) using as input the reference sequence. Hydropathy plots were also used to evaluate the effect of residue mutations. Sequences with a propensity to partition into the lipid bilayer were identified using interfacial settings and pH = 5.0.

## Contact prediction on the Cys-rich region

Contact prediction was performed using RaptorX (*Wang et al., 2017*; *Teppa et al., 2020*). RaptorX integrates evolutionary coupling and sequence conservation information through an ultra-deep neural network formed by two deep residual neural networks. RaptorX predicts pairs of residues, whose mutations have arisen simultaneously during evolution.

## Structural models and molecular dynamic simulation studies

The HBV surface protein sequence was taken from the UniProt database, with accession number P03138. Secondary structure prediction was performed with Jpred (*Cole et al., 2008*). The S protein region 294–317 was modeled using MODELLER (*Sali and Blundell, 1993*). The template crystal structure of the Newcastle disease virus fusion protein (PDB code: 1G5G) was retrieved from the PDB database (*Berman et al., 2000*). Sequence alignment was generated with Clustal X (*Larkin et al., 2007*). The model evaluation was conducted using the Ramachandran plot (*Ramachandran et al., 1963*). The model of the wild-type sequence was further used to create two structural models with mutations using UCSF Chimera package (*Pettersen et al., 2004*). One model contains the double mutations T303C/G308C, which may create an extra disulfide bond. The overall effect of those mutations would be to 'shift' the disulfide bridge of two amino acids toward the turn of the β-hairpin motif. After mutations, the models were energy minimized by applying Molecular Modelling Toolkit (MMTK) with Amber parameters for standard residues and 100 steepest descent minimization steps with a step size of 0.02 Å. To investigate the stability of the disulfide bonds, MD simulations of the three models were carried out by GROMACS version 2020 (*Abraham et al., 2015*) in conjunction with OPLS-AA/L all-atom force field. The models were immersed in the cubic boxes filled with water molecules with a minimal distance of 1.0 nm between the peptide surface and the box. Each system was equilibrated to the desired temperature through a stepwise heating protocol in NVT ensemble followed by 100.0 ps equilibration in NPT ensemble with position restraints on the protein molecule. The final productive MD was performed for each system for 10 ns under periodic boundary conditions without any restraints on the protein with a time step of 2 fs at constant pressure (1 atm) and temperature (300 K). Coordinates were saved every 10 ps, yielding 1000 frames per MD trajectory. All the frames were further investigated to differentiate between allosteric and structurally stabilizing disulfides. Disulfide bonds were classified based on the five relevant torsion angles ($\chi_1$, $\chi_2$, $\chi_3$, $\chi_{2'}$, and $\chi_{1'}$) (see *Figure 4—figure supplement 2*), disulfides being treated as symmetrical. In this system, 20 conformational categories are possible (*Marques et al., 2010*; *Schmidt and Hogg, 2007*; *Schmidt et al., 2006*). The three central angles ($\chi_2$, $\chi_3$, and $\chi_{2'}$) define the basic shapes: Spiral, Hook, and Staple (*Eklund et al., 1984*). $\chi_3$ defines the orientational

motif: left-handed (LH) or right-handed (RH) if the sign is negative or positive, respectively (*Eklund et al., 1984*). $\chi_1$ and $\chi_1'$ determine the signs of the nomenclature (*Qi and Grishin, 2005*).

## In vivo experiments

All experiments were performed in accordance with the European Union guidelines for approval of the protocols by the local ethics committee (Authorization Agreement C2EA-15, 'Comité Rhône-Alpes d'Ethique pour l'Expérimentation Animale', Lyon, France - APAFIS#27316–2020060810332115 v4). Primary human hepatocytes (PHH; Corning, BD Gentest) were intrasplenically injected into NFRG mice (*Azuma et al., 2007*), a triple mutant mouse knocked out for fumarylacetoacetate hydrolase (fah$^{-/-}$), recombinase activating gene 2 (rag2$^{-/-}$), and interleukin two receptor gamma chain (IL2rg$^{-/-}$), 48 hr after adeno-uPA conditioning (*Bissig et al., 2010*; *Calattini et al., 2015*). Mice were subjected to 2-(2-nitro-4-trifluoro-methylbenzoyl)1,3-cyclohexedione (NTBC) cycling during the liver repopulation process, as described previously (*Calattini et al., 2015*). Mice with HSA levels >15 mg/ml, as determined using a Cobas C501 analyzer (Roche Applied Science), were inoculated with virus preparations by intra-peritoneal injections. Sera were collected at different time points before and after infection. Mice were sacrificed 6 weeks post-infection.

## Statistical analysis

Statistical analyses were performed using GraphPad Prism version 5.02 for Windows, GraphPad Software (San Diego, California, USA). The Mann-Whitney or Wilcoxon tests were used for statistical comparisons. A p-value of 0.05 or less was considered as significant. When applicable, data are presented as mean ± standard deviation and results of the statistical analysis are shown as follows: ns, not significant ($p > 0.05$); *$p < 0.05$; **$p < 0.01$; and ***$p < 0.001$.

## Acknowledgements

We are grateful to Camille Sureau for the HBV GP expression constructs and for sharing the HDV infection assay. We thank Solène Denolly for helpful discussions.

We thank the 'Plateforme de Thérapie Génique' in Nantes (France) for the production of the in vivo-certified lots of adeno-uPA vectors. We thank Jean-François Henry, Nadine Aguilera and Tiphaine Dorel from the animal facility (PBES, Plateau de Biologie Experimental de la Souris, UMS3444/CNRS, US8/Inserm, ENS de Lyon, UCBL), and Véronique Pierre for their technical help in the handling of mice. We acknowledge the contribution of ANIRA-Genetic Analysis and the PLATIM-Microscopy facilities of SFR Biosciences (UMS3444/CNRS, US8/Inserm, ENS de Lyon, UCBL) for image quantifications, technical assistance, and support. We thank Didier Décimo for support with the BSL3 facility. We thank Omran Allatif for guidance with the statistical analysis. This work was supported by the French 'Agence Nationale de la Recherche sur le SIDA et les Hépatites Virales | Maladie Infectieuses Emergentes' (ANRS|MIE, grants ECTZ71388, ECTZ160643, ECTZ160726, ECTZ38814, and ECTZ41733), the Foundation FINOVI, the 2017 Call for Joriss Projects, the LabEx Ecofect (ANR-11-LABX-0048) of the 'Université de Lyon', within the program 'Investissements d'Avenir' (ANR-11-IDEX-0007) operated by the French National Research Agency (ANR), and the LabEX CALSIMLAB (ANR-11-LABX-0037–01 and ANR-11-IDEX-0004–02) of Sorbonne Université.

## Additional information

### Funding

| Funder | Grant reference number | Author |
| --- | --- | --- |
| LabEx Ecofect of the Université de Lyon | ANR-11-LABX-0048 | François-Loïc Cosset |
| ANR | ANR-11-IDEX-0007 | François-Loïc Cosset |
| LabEX CALSIMLAB of Sorbonne Université | ANR-11-LABX-0037–01 | Alessandra Carbone |
| ANR | ANR-11-IDEX-0004–02 | Alessandra Carbone |
| ANRS|MIE | ECTZ71388 | François-Loïc Cosset |

| | | |
|---|---|---|
| | | Anja Böckmann |
| ANRS\|MIE | ECTZ160643 | Alessandra Carbone<br>François-Loïc Cosset |
| ANRS\|MIE | ECTZ160726 | Alessandra Carbone<br>François-Loïc Cosset<br>Natalia Freitas |
| ANRS\|MIE | ECTZ38814 | Alessandra Carbone<br>François-Loïc Cosset<br>Elin Teppa |
| ANRS\|MIE | ECTZ41733 | Alessandra Carbone<br>François-Loïc Cosset<br>Christophe Combet |
| ANRS\|MIE | ECTZ123278 | François-Loïc Cosset |
| ANRS\|MIE | ECTZ119828 | François-Loïc Cosset |
| Foundation FINOVI | | François-Loïc Cosset |
| 2017 Call for Joriss Projects | | François-Loïc Cosset |

The funders had no role in study design, data collection and interpretation, or the decision to submit the work for publication.

## Author contributions

Jimena Pérez-Vargas, Conceptualization, Data curation, Formal analysis, Supervision, Validation, Investigation, Visualization, Methodology, Writing - original draft, Writing - review and editing; Elin Teppa, Conceptualization, Data curation, Software, Formal analysis, Validation, Investigation, Visualization, Methodology, Writing - original draft, Writing - review and editing; Fouzia Amirache, Rémi Pereira de Oliveira, Floriane Fusil, Investigation, Methodology; Bertrand Boson, Data curation, Formal analysis, Validation, Investigation, Methodology, Writing - review and editing; Christophe Combet, Anja Böckmann, Resources, Software; Natalia Freitas, Conceptualization, Data curation, Formal analysis, Supervision, Validation, Investigation, Visualization, Methodology, Writing - review and editing; Alessandra Carbone, Conceptualization, Resources, Data curation, Software, Formal analysis, Supervision, Funding acquisition, Validation, Visualization, Methodology, Writing - original draft; François-Loïc Cosset, Conceptualization, Resources, Data curation, Formal analysis, Supervision, Funding acquisition, Validation, Visualization, Writing - original draft, Project administration, Writing - review and editing

## Author ORCIDs

Elin Teppa ⓘD https://orcid.org/0000-0002-0691-9654
Fouzia Amirache ⓘD https://orcid.org/0000-0002-6453-3717
Bertrand Boson ⓘD https://orcid.org/0000-0003-1920-5172
Rémi Pereira de Oliveira ⓘD https://orcid.org/0000-0002-3005-1277
Christophe Combet ⓘD https://orcid.org/0000-0002-7348-3520
Anja Böckmann ⓘD https://orcid.org/0000-0001-8149-7941
Natalia Freitas ⓘD https://orcid.org/0000-0001-8821-561X
Alessandra Carbone ⓘD https://orcid.org/0000-0003-2098-5743
François-Loïc Cosset ⓘD https://orcid.org/0000-0001-8842-3726

## Ethics

Animal experimentation: All experiments were performed in accordance with the European Union guidelines for approval of the protocols by the local ethics committee (Authorization Agreement C2EA-15, "Comité Rhône-Alpes d'Ethique pour l'Expérimentation Animale", Lyon, France - APA-FIS#27316-2020060810332115 v4).

## Decision letter and Author response

Decision letter https://doi.org/10.7554/eLife.64507.sa1

Author response https://doi.org/10.7554/eLife.64507.sa2

## Additional files

### Supplementary files

• Supplementary file 1. Oligonucleotide sequences used for shRNAs and mutagenesis. The sequences correspond to the oligonucleotides used to generate the lentiviral vectors carrying shRNA against the indicated protein disulfide isomerases (PDIs) in *Figure 7* or the hepatitis B virus glycoprotein (HBV GP) mutants described in *Figure 2* (preS1 and preS2 mutants) and in *Figures 4* and *5* (CSD mutants).

• Transparent reporting form

### Data availability

All data generated or analysed during this study are included in the manuscript and supporting files. Source data files have been provided for figures 1-3, 5-7 and 9.

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
