## [Decision Letter]

**Acceptance summary:**

Hepatitis B virus (HBV) is an enveloped virus that enters cells by membrane fusion. Though studied extensively, the mechanisms underlying HBV entry/fusion have not been established. Previous work has indicated that low pH does not act as cellular cue for HBV fusion, as is the case for many viruses. This paper describes modelling studies, and supporting experiments, that identify a potential fusion domain and support a fusion mechanism involving disulphide exchange. The work provides crucial insights to the molecular mechanism through which this important virus infects cells.

**Decision letter after peer review:**

Thank you for submitting your article "A fusion peptide in preS1 and the protein-disulfide isomerase ERp57 are involved in HBV membrane fusion process" for consideration by *eLife*. Your article has been reviewed by 3 peer reviewers, one of whom is a member of our Board of Reviewing Editors, and the evaluation has been overseen by José Faraldo-Gómez as the Senior Editor. The reviewers have opted to remain anonymous.

The reviewers have discussed the reviews with one another and the Reviewing Editor has drafted this decision to help you prepare a revised submission.

As the editors have judged that your manuscript is of interest, but also that additional experiments are required before it can be again considered for publication, we would like to draw your attention to changes in our revision policy that we have made in response to COVID-19 (https://elifesciences.org/articles/57162). First, because many researchers have temporarily lost access to the labs, we will give authors as much time as they need to submit revised manuscripts. We are also offering, if you choose, to post the manuscript to bioRxiv (if it is not already there) along with this decision letter and a formal designation that the manuscript is "in revision at *eLife*". Please let us know if you would like to pursue this option. (If your work is more suitable for medRxiv, you will need to post the preprint yourself, as the mechanisms for us to do so are still in development.)

Summary

Hepatitis B virus (HBV) is an enveloped virus that gains entry to cells by membrane fusion. Though studied extensively, the mechanisms underlying HBV entry/fusion have not been established. Previous work has indicated that low pH appears not to act as cellular cue for HBV fusion, as is the case for many viruses. This paper describes modelling studies, and supporting experiments, that identify a potential fusion domain and support a mechanism involving disulphide exchange in fusion. The work provides crucial insights to the molecular mechanism through which this important virus infects cells.

Essential revisions

The reviewers consider the work is interesting and potentially publishable in *eLife*; however, they all raise significant concerns that should be addressed. These include:

1) Analysis of the pre-fusion and reorganised disulphides.

2) Information of the location of ERp57 and how this relates to the likely site of fusion.

3) Assessment of the role of HSPGs in fusion.

4) Characterisation of the HDV particles.

5) Quantitation of comment on the cell-cell fusion assay.

The full reviews are provided below. We strongly urge you to consider these carefully and address the points raised either through the inclusion of additional data or modifications to the text of the manuscript.

*Reviewer #1:*

Using compounds that inhibit PDIs and RNAi, the authors conclude that ERp57 is likely the primary PDI involved in triggering HBV fusion. As the name suggests ERp57 is primarily associated with the ER. Is there evidence of its expression on the cell surface, which would presumably be necessary to support the cell-cell fusion assays used in this study, or in the endocytic pathway? Although there is evidence of some viruses (e.g. polyoma) being able to gain access to the ER following endocytosis, I am not aware this is the case for HBV.

*Reviewer #2:*

1. The fact that the specific HBV receptor NTCP is not required in the fusion assays is surprising and a potentially important finding. This may indicate that the cell fusion assays show a phenomenon later, after virus entry into vesicles but may also indicate unspecific interactions driven by the HBV surface proteins, exposed in a non-physiological, artificial manner. For proving specificity of the cell fusion assays the authors should control if HSPGs are required.

2. If the exposure of the three HBV surface proteins is sufficient for efficient cell-to-cell fusion, I am wondering why there are no in vivo data, supporting that HBV infected hepatocytes fuse with their surrounding cells, which – according to the data of the manuscript – must not even be hepatocytes. Either this is a rare event or in vivo exposure of the HBV surface proteins is fundamentally different.

3. The additional disulfide bond in mt T301C-G310C should be experimentally shown. The conformational rearrangements mentioned in the manuscript (e.g. p8) should be experimentally verified. This could be achieved e.g. by immune precipitations using mAbs for showing accessibility of the AGL and preS1 exposure. This could be achieved e.g. by immune precipitations using mAbs for showing accessibility of the AGL and preS1 exposure.

4. The manuscript shows a large number of indirect result figures (i.e. normalized to 100% of three experiments), without giving the original data. This includes pictures and quantifications of cell-to-cell fusions (see 2.)

5. Conformational analyses i.e. proper folding of the HBV surface proteins was based on HD virion formation. This was tested by RTqPCR in the supernatant after 0.45-micron filtration up to 9 days post transfection. I see no reason why the PCR should detect HDV RNA encapsidated in HDV particles only, which should be verified e.g. by ultracentrifugation, immunoprecipitation followed by PCR etc.

6. Figure 2I shows that mt G53A, which allows HDV infection similar to wt HDV, exposes practically only LHBs and very little (if at all) SHBs, which is clearly visible in Figure 1C. This implies that despite of the statement that the mutants do not interfere with virion formation, they may have a significant impact on surface protein folding and exposure on the cell surface.

7. When using inhibitors of e.g. protein disulfide isomerases, no toxicity data were shown.

*Reviewer #3:*

1) E.g. Alkylation of free sulfhydryls and reduction of disulfides could be done with purified virions, the agents dialysed away, and the chemically modified virions could be used for infectivity studies.

2) E.g. non-reducing gels of S or L protein expressing cells could indicate the presence of disulfide bonds. Combined with the use of cysteine mutants the pattern of shifts will indicate formation or absence of disulfides, and allow conclusions on their positioning. Combinations of alkylation, reduction, alkylation with a different agent followed by mass spectrometry may indeed directly show which sulfhydryls are in a disulfide bond or free. This analysis may also be done prior to and post fusion of cells, which may in fact show the isomerization reaction.

3) It would be important to demonstrate where ERp57 functions to allow fusion. Is it at the plasma membrane or within virus-containing endosomes? Do they physically interact? Perhaps an in vitro isomerization assay would increase the confidence that ERp57 can directly act on the viral protein.

---

## [Author Response]

Essential revisionsThe reviewers consider the work is interesting and potentially publishable in ; however, they all raise significant concerns that should be addressed. These include:1) Analysis of the pre-fusion and reorganised disulphides.2) Information of the location of ERp57 and how this relates to the likely site of fusion.3) Assessment of the role of HSPGs in fusion.4) Characterisation of the HDV particles.5) Quantitation of comment on the cell-cell fusion assay.The full reviews are provided below. We strongly urge you to consider these carefully and address the points raised either through the inclusion of additional data or modifications to the text of the manuscript.Reviewer #1:Using compounds that inhibit PDIs and RNAi, the authors conclude that ERp57 is likely the primary PDI involved in triggering HBV fusion. As the name suggests ERp57 is primarily associated with the ER. Is there evidence of its expression on the cell surface, which would presumably be necessary to support the cell-cell fusion assays used in this study, or in the endocytic pathway? Although there is evidence of some viruses (e.g. polyoma) being able to gain access to the ER following endocytosis, I am not aware this is the case for HBV.

We thank this Reviewer for giving us the opportunity to experimentally address this important question. While PDIs are generally abundantly located in the ER, their capacity to traffic beyond the ER in the secretory pathway has been documented and seems to depend both on specific PDI members and cell types. For example, in addition to the ER, ERp57 has been detected at the cell surface as well as in the nucleus, plasma membrane rafts, and cytosol, as discussed in Turano et al. (2002) and references herein. Of note, ERp46, ERp57 and ERp72 were selected in our study because they can be detected at the cell surface. These considerations are now better discussed in the revised version of our manuscript (pages 9-10 and 13-14).

Noteworthy, we have experimentally addressed this important concern in our Huh7 cells-based assays.

First, we show in revised Figure 7A a FACS analysis of Huh7-NTCP cells that reveal a significant albeit low expression of ERp57 at the cell surface, which supports our results of cell-cell fusion.

Furthermore, we also show in a new Figure 8 some immuno-fluorescence images by confocal microscopy analysis of Huh7-NTCP cells stained with antibodies against ERp57 and Rab5 (early endosomes), Rab7 (late endosomes), Rab11 (recycling endosomes) or Lamp1 (lysosomes). The quantifications of these results show that ERp57 can be detected in late endosomes but poorly in the other above-tested locations. We believe that these results are meaningful since previous reports showed that HBV infection of HepaRG cells depends on Rab5 and Rab7 (Macovei et al., 2013), which are GTPases involved in the biogenesis of endosomes, and that the epidermal growth factor receptor (EGFr) is a host-entry cofactor that interacts with NTCP and mediates HBV internalization (Iwamoto et al., 2019).

Altogether, we confirm that in addition to the ER, ERp57 can be found at locations compatible for both cell-cell fusion and cell-free entry by internalization. The Results and Discussion sections have been modified accordingly (pages 9-10, pages 13-14).

Reviewer #2:1. The fact that the specific HBV receptor NTCP is not required in the fusion assays is surprising and a potentially important finding. This may indicate that the cell fusion assays show a phenomenon later, after virus entry into vesicles but may also indicate unspecific interactions driven by the HBV surface proteins, exposed in a non-physiological, artificial manner. For proving specificity of the cell fusion assays the authors should control if HSPGs are required.

We thank this Reviewer for giving us the opportunity to address this important point. Indeed, we were first surprised by these results, although for other viruses such as e.g., influenza or HCV, binding to their respective cell-free entry receptors is not a requirement for both cell-cell fusion (Lin and Cannon, 2002) and liposome fusion (Lavillette et al., 2006) assays triggered by low pH treatment. Thus, while it is clear that cell-cell fusion does not recapitulate *per se* all the events required to promote cell entry of viral particles, such an assay bypasses the step of internalization that subsequently allows membrane fusion in endosomes, which are the sites of entry for these abovementioned viruses and as proposed for HBV (Macovei et al., 2013; Iwamoto et al., 2019).

Noteworthy, we have experimentally addressed this important concern further.

First, we show in revised Figure 1B the results of cell-cell fusion assays performed in the presence of an HSPG blocker, i.e. heparin, which was present throughout the co-culture of donor cells (expressing HBV glycoproteins) and acceptor cells (expressing, or not, NTCP). While the applied doses of heparin could prevent cell-free entry, as shown in Schulze et al. (2007), our results indicate that HSPG blocking does not inhibit cell-cell fusion.

Furthermore, we also show in revised Figure 1C the results of cell-cell fusion assays performed with Chinese ovary cells (CHO) donor cells co-cultured with either CHO cells or CHO-pgsB618 cells. While both cell types do not express NTCP, the latter one does not express HSPGs (Richard et al., 1995). Our results indicate that cell-cell fusion could be detected for either cell type to the same extent as for Huh7 cells.

Overall, these results show that HBV cell-cell fusion requires neither NCTP nor HSPG, which is now described page 5.

As for the comment of this Reviewer that our cell-cell fusion results “may also indicate unspecific interactions driven by the HBV surface proteins, exposed in a non-physiological, artificial manner”, we believe first that previous results studies using the above-mentioned enveloped viruses (influenza and HCV as well as e.g., Togaviruses, Flaviviruses, Rhabdoviruses, Bunyaviruses, and Arenaviruses – see Earp et al., 2004) are in line with our results obtained with HBV and, second, that the cell fusion assays indeed reflect a phenomenon at a late stage of entry, after virus binding and transport into vesicles, which becomes receptor-independent. We discuss these aspects in the revised Discussion (page 11).

2. If the exposure of the three HBV surface proteins is sufficient for efficient cell-to-cell fusion, I am wondering why there are no data, supporting that HBV infected hepatocytes fuse with their surrounding cells, which according to the data of the manuscript must not even be hepatocytes. Either this is a rare event or exposure of the HBV surface proteins is fundamentally different.

Indeed, to the best of our knowledge but also in agreement with our IF analysis of livers from HBV-infected huHep mice (Perez-Vargas et al., 2019), there seems to be no data supporting HBV GP mediated cell-cell fusion in vivo. One possibility is that it may be difficult to achieve a sufficient level of GP expression at the cell surface of HBV-infected cells, in contrast to transfected cells as in our cellcell fusion assay. A second possibility is that the expression level and localization of ERp57 could be differentially regulated in vivo *vs.* in vitro. Note that unlike other enveloped viruses whose GP expression readily fuses neighboring cells and forms multinucleated giant cells (e.g., Measles), cell-cell fusion with HBV GPs may result in syncytia that hardly harbor over two nuclei, which is very difficult to observe in liver slices from in vivo experiments.

3. The additional disulfide bond in mt T301C-G310C should be experimentally shown. The conformational rearrangements mentioned in the manuscript (e.g. p8) should be experimentally verified. This could be achieved e.g. by immune precipitations using mAbs for showing accessibility of the AGL and preS1 exposure. This could be achieved e.g. by immune precipitations using mAbs for showing accessibility of the AGL and preS1 exposure.

We separately address either concern below:

Concerning “The additional disulfide bond in mt T301C-G310C* should be experimentally shown”

As an attempt to demonstrate that the amino acids substitutions T303C and G308C (TG/CC mutant) induce the formation of an extra disulfide, in addition to the predicted cross-strand disulfide (CDS) between cysteines C301 and C310, we first labeled wt or TG/CC GPs with maleimide-containing reagents: 4-acetamido-40-maleimidylstilbene-2,2’-disulfonic acid (AMS) or 2 kDa PEG-maleimide polymer (mPEG) following treatment with TCEP (as disulfide bond reducer). Both alkylating agents, which specifically bind free thiol groups, should induce detectable electrophoretic migration shifts of ~0.5 kDa and 2.2 kDa per free thiol group, respectively, on HBV GPs. Since HBV GP contains 14 cysteine residues, all located in the S domain, the size of TCEP reduced wt GPs should be increased by ~7 kDa after treatment with AMS and by ~31 kDa after labeling with mPEG. Thus, a 1kDa (for AMS) or 4.4kDa (for mPEG) difference is expected between the wt and the TG/CC mutant, which contains 2 additional cysteines. Second, we reasoned that alkylation without prior treatment with TCEP should lead to formation of thioether bonds between AMS or mPEG and potential free thiols. Thus, should T303C and G308C form a disulfide, we would expect no differences in the sizes of AMS- or mPEG-treated protein samples between the wt and TG/CC mutant GPs.

To test either hypothesis, we performed reduction and/or alkylation of protein lysates from cells expressing wt or TG/CC mutant GPs and recovered by TCA/acetone precipitation, prior to SDS-PAGE and Western blot analysis. As shown in Author response image 1, while HBV GPs were readily detectable after β-mercaptoethanol (A) or TCEP (B) reduction, further alkylation of HBV GPs with AMS prevented their recognition by the Murex antibody (B) that targets the AGL of HBsAg. Aiming to circumvent this, we blotted the membranes with an anti-preS1 antibody instead, leading to detection of the L protein only (A). Treatment with AMS increased the HBV envelope protein’s mass of both wt and TG/CC mutant GPs, though the expected 1kDa difference between the wt and the TG/CC mutant was not clear (B). Treatment of HBV GPs with mPEG rendered them undetectable by the two antibodies used (not shown). Likewise, direct treatment of HBV GPs with AMS or mPEG rendered them undetectable in Western blot analysis (not shown), which prevented us to show the additional disulfide bond.

**Author response image 1. respfig1:** HBsAg characterization. (A) Lysates of cells transfected with wild type HBV LMS (WT) HBC LMS mutant. T303C/G308C (TG/CC) were treated with β-mercaptoethanol and denatured were analyzed by immunoblotting using Murex (left) ou preS1 (right) antibodies. (B) Lysates of cells transfected with wild type HBV LMS (WT) HBC LMS mutant T303C/G308C (TG/CC) were treated with TCEP, which reduces disulfide bonds, and AMS, which alkylates free thiols, as indicated. Samples were then analyzed by immunoblotting using Murex (left) ou preS1 (right) antibodies. The results show that after treatment with AMS, Murex antibodies did not recognize any more HBV GPs. Using preS1 antibodies, we detected a 7kDa shift in the L protein due to reaction of AMS with free thiols but we were unable to detect 1 kDa difference between wt and the double cysteine TG/CC mutant.

Concerning “The conformational rearrangements mentioned in the manuscript (e.g. p8) should be experimentally verified”

Previous studies showed that the reactivity of HBV particles to antibodies against HBsAg was dependent on disulfide bonds (Vyas et al., 1972) and that the antigenicity of sub-viral particles carrying AGL cysteine mutations was drastically affected (Mangold et al., 1995). Moreover, the individual contribution of each cysteine of the antigenic loop (AGL) to the production and infectivity of HDV was extensively analyzed (Abou-Jaoudé and Sureau, 2007; Salisse and Sureau, 2009). Using commercial immunoassays that utilize monoclonal antibodies directed to the immunodominant “a” determinant, it was confirmed that mutation of any of the tested AGL cysteines drastically reduced the antigenicity of HDV particles (see Figure 4 from Salisse and Sureau, 2009). Though, the reduced antigenicity and infectivity was not always accompanied by inhibition of HDV assembly and secretion (AbouJaoudé and Sureau, 2007). Nevertheless, the structure of the AGL seemed to be correlated with infectivity, and thus was defined as the second determinant for HBV/HDV cell entry. Importantly, in addition to the cysteines, several non-cysteine residues were recognized as important for both structure and infectivity of the AGL. Yet, neither T140A (corresponding to T303 in our study) nor G145A (corresponding to G308 in our study) were identified as critical for both the structure of AGL and HDV production and infectivity (Salisse and Sureau, 2009). Furthermore, mutations at these positions (T140 and G145) have been detected in naturally occurring replication competent surface antigen variants of HBV. The vaccine escape mutation G145R has a high impact on AGL structure without affecting infectivity. Indeed, the mutation figure from Salisse and Sureau, 2009 (Figure 4). Specific antigenicity and was shown to confer increased infectivity of HDV particles bearing substitutions of serine for cysteine cell-binding capacity via residues or alanine for noncysteine residues in the AGL of HBsAg. interaction with HSPGs.

Altogether, these data indicated a partial correlation between the three-dimensional structure of the AGL inferred by immunoassays, assembly and secretion, and particle attachment and infectivity (Sureau and Salisse, 2013).

On this ground, to address the comment of this Reviewer, we attempted to identify the changes induced by the introduction of T140C (T303C) and G145C (G308C) mutations in the AGL determinant (T303C-G38C mutant). First, we examined HDV particles for antigenicity using the Murex HBsAg version 3 ELISA and a chemiluminescence immunoassay (CLIA) for detection of HBsAg antigen (see Author response table 1 below).

**Author response table 1. resptable1:** Antigenicity of HDV particles. RT-qPCR assay quantifying HDV RNA in the supernatants of producer cells was used to normalize preparations of viral particles prior to being subjected to ELISA specific for the AGL determinant. The results are presented as fold-change relative to that of the wt. GE: HDV RNA genome equivalents; no-GP: supernatants of Huh7 cells transfected with a trimer of HDV cDNA (pSVL-D3) and an empty plasmid; wt-LMS: supernatants of Huh7 cells transfected with pSVL-D3 and HBV wt LMS envelope proteins; Wt-S: HBV S only; LMS(T303C): HBV LMS envelope proteins bearing a T140C mutation; LMS(TG/CC): HBV LMS envelope proteins bearing T140C and G145C mutations.

MUREX	1E4 GE	1E5 GE	CLIA	1E4 GE	1E5 GE
no-GP	neg	neg	no-GP	neg	neg
wt-LMS	1.00	1.00	wt-LMS	1.00	1.00
wt-S	1.20	1.13	wt-S	1.64	1.43
LMS(T303C)	0.91	0.97	LMS(T303C)	neg	neg
LMS(TG/CC)	neg	neg	LMS(TG/CC)	neg	neg

Whereas HDV particles assembled with wt HBV LMS or with S protein only were equally detected by the two ELISA assays, we found disparities regarding the mutant HDV particles. Indeed, as shown in this Table, the CLIA assay failed to detect any of the mutants, although the T140C (T303C) mutant was readily detected by the Murex immunoassay kit. This confirmed that assessment of AGL exposure is strictly dependent on the anti-HBsAg antibodies used in the immunoassay, which can lead to misleading results. These results are unfortunately in agreement with previous studies that demonstrated that amino acid substitutions in the AGL determinant of HBsAg may account for false negative results in immunoassays, independently of the virus load in sera of infected patients, and, as shown in this study, independently of HDV RNA viral load in the supernatants.

Overall, due to antigenic loss of cysteine mutants, the above results did not allow us to design an assay that can detect the additional disulfide bond in the T303C-G308C mutant nor the block of conformational rearrangements that is suggested by the phenotype of this mutant. We hope that this Reviewer understands this severe limitation that precluded to fully address his/her comment. Accordingly, we have introduced a modification in the revised Discussed to convey this point (page 13).

Nevertheless, we showed that the mutant HDV particles bind Huh7 cells as efficiently as HDV assembled with wt HBV LMS (Figure 5A), allowing us to infer that the changes induced by mutations in the antigenic structure of AGL and/or 294-317 epitope do not impair the ability of HDV particles to attach to the cell surface. Furthermore, we recall that we confirmed preS1 exposure on HDV by testing the ability of mutant particles to infect Huh7-NTCP cells. Indeed, the two single cysteine mutants with altered antigenicity were as infectious as HDV particles assembled with wt HBV LMS (Figure 5B).

4. The manuscript shows a large number of indirect result figures (i.e. normalized to 100% of three experiments), without giving the original data. This includes pictures and quantifications of cell-to-cell fusions (see 2.)

The quantification of cell-cell fusion was done by measuring the luciferase activity induced by fusion between the donor and indicator cells, as explained in the Materials and methods. The absolute values of luciferase activity could show some variations between one experiment to another, owing to cell growth differences in the co-cultures. Thus, in order to address the statistical significance of the different experiments, we chose to express the results as percentage of the WT condition at pH7. Yet, we now show in a new Figure 1—figure supplement 1 the crude results of three independent experiments of cell-cell fusion (used to generate Figure 1A) that are expressed as ratio of values to that of transfection with the empty plasmid.

5. Conformational analyses i.e. proper folding of the HBV surface proteins was based on HD virion formation. This was tested by RTqPCR in the supernatant after 0.45-micron filtration up to 9 days post transfection. I see no reason why the PCR should detect HDV RNA encapsidated in HDV particles only, which should be verified e.g. by ultracentrifugation, immunoprecipitation followed by PCR etc.

HDV RNA replication can be simply initiated by transfection of cell lines with cDNA constructs expressing either genomic or antigenomic HDV RNA as long as they encode a functional S-HDAg. Though, for HDV assembly and release to occur, these cells must also co-express the large form of the δ antigen (L-HDAg), which results from an editing event on the virus antigenome, and envelope proteins from the helper virus, i.e., the HBV surface glycoproteins (HBsAgs). Before standardization of RT-qPCR assays for absolute quantification of HDV genomes, virus replication and egress were conventionally assessed by Northern blot analysis with radioactive probes. As demonstrated before by Camille Sureau’s group, HDV RNA cannot be detected by Northern blotting in the supernatants of Huh7 cells co-transfected with pSVL-D3 (replication competent HDV cDNA construct) and a plasmid devoid of the sequences for the HBsAgs (see e.g., Figure 4 in Julithe et al., 2014). Northern blotting has been gradually replaced by RT-PCR assays, which are more sensitive, to quantify HDV replication following infection as well as HDV titers (total physical particles) released from transfected cells or circulating in human sera. A side-by-side comparison of HDV infectivity both by Northern blot and qPCR validated the latter as an accurate and sensitive method to deduce the number of HDV equivalents present in a sample (see e.g., Table 3 in Gudima et al., 2007). Because within the HDV virion there is one copy of the genome that is associated with 70-200 copies of HDAg, the HDV RNA levels correlate with the number of secreted HDV particles. Moreover, since the HDV RNPs require HBsAgs for assembly and egress, HDV titers are commonly determined by qPCR and expressed as genome equivalents (GE) per ml, representing the total number of physical particles.

As depicted in the Author response image 2, the HDV RNA detected in the producer cell supernatant in a transfection assay lacking the sequences for HBsAg (noGP) represents less than 1% of the HDV genome equivalents detected in the supernatants of cells producing HDV particles. This likely corresponds to HDV RNAs released from dead cells, independently of whether the RNA was extracted from crude supernatants or from supernatants that have been concentrated by ultracentrifugation through a

30% sucrose cushion.

Consistently, HBsAgs but also HDAgs were undetectable in the noGP fraction by Western blot, contrasting with detection of both proteins when more that 1E+07 GE of HDV RNA was loaded.

**Author response image 2. respfig2:** HDV particles characterization. (A) HDV RNA titers (GE/μl) in crude supernatants from producer cells co-transfected with pSVL-D3 and with i) empty plasmid (noGP), ii) wild type HBV LMS GPs (Wt), iii) HBV S only (S), or iv) HBV LMS mutant T303C (Cys mutant), and in 100-fold concentrated virus samples subjected to ultracentrifugation through 30% sucrose cushion (UC). (B) Western blot analysis of 100-fold concentrated particles (1E7 GE) from ultrafiltrated crude supernatants or pellets after ultracentrifugation with anti-HBsAg antibody (Murex) and rabbit polyclonal serum against HDAg.

6. Figure 2I shows that mt G53A, which allows HDV infection similar to wt HDV, exposes practically only LHBs and very little (if at all) SHBs, which is clearly visible in 1C. This implies that despite of the statement that the mutants do not interfere with virion formation, they may have a significant impact on surface protein folding and exposure on the cell surface.

In the repetitions of this experiment, while the results of the specific experiment displayed in the previous version of Figure 2I exhibited a poorer expression of the mutant G53A as compared to WT, the other experiments showed that L, M and S GPs are expressed from mutant G53A at levels similar to the wt. Yet, we had provided the quantification of these experiments in previous version of supplemental figure 4A, which indicated that all mutants in preS1 were expressed at levels similar to WT. We have now introduced this quantitative analysis in the revised Figure 2I, 2J to better convey this notion and we display in revised Figure 2—figure supplement 2 some western blots for illustrations.

7. When using inhibitors of e.g. protein disulfide isomerases, no toxicity data were shown.

We apologize for this omission. The results of toxicity data for all inhibitors are now shown in the new Figure 1—figure supplement 2.

Reviewer #3:1) E.g. Alkylation of free sulfhydryls and reduction of disulfides could be done with purified virions, the agents dialysed away, and the chemically modified virions could be used for infectivity studies.

In response to this Reviewer’s suggestion, we would like to recall a previous study of Camille Sureau’s group in 2007 that identified the cysteines of the S domain essential for HDV infectivity, all being located in the AGL, and that extensively analyzed the effect of disulfide bond reducing agents (TCEP and DTT) and alkylators (DTNB, AMS, MTSET and M135) on viral entry (Abou-Jaoudé and Sureau, 2007). First, and in support of our findings that the region 294-317 of the LMS sequence contains a cross-strand disulfide (CDS) between C301 and C310 (Figure 4, our manuscript), the authors showed that substitutions of either cysteine had a pronounced inhibitory effect on infectivity (Figure 3). Second, they also analyzed the effect of reducing or alkylating agents on HDV entry. The inhibitors were added at the time of infection (co-inoculation) or added at 1 day post-inoculation (Figure 5, in Abou-Jaoudé and Sureau, 2007). Similar to the results we obtained with the alkylator DTNB (Figure 3 and Figure 3—figure supplement 1, our manuscript), it was clearly demonstrated that HDV infection is inhibited when the drugs are present during the period of virus-cell exposure but have no effect if added after virus entry. Finally, to further address the importance of thiols/disulfide exchange at the surface of HDV and its implication at viral entry, Abou-Jaoudé and Sureau (2007) pre-treated HDV particles with different doses of AMS (alkylator) or TCEP (reducer) prior to infection. Pre-treatment of HDV particles, with either of the inhibitors, caused HDV loss of infectivity (Figure 6, in Abou-Jaoudé and Sureau, 2007). These previous results are in favor of the hypothesis that not all the cysteines are engaged in disulfide bridging and that free-thiol groups are necessary to catalyze disulfide reduction and likely isomerization events of disulfide bonds events during virus entry. Consistent with this view, we showed that stabilizing the loop containing the putative CSD bond inhibited virus entry and fusion (Figure 5, our manuscript).

Hence, we believe that conducting the experiments requested would merely confirm the previous observations described in Abou-Jaoudé and Sureau (2007). Instead, bringing new insights into the mechanism of HBV membrane fusion, we identified ERp57, a member of PDI-family, as a host factor critically involved in triggering HBV fusion and infection.

2) E.g. non-reducing gels of S or L protein expressing cells could indicate the presence of disulfide bonds. Combined with the use of cysteine mutants the pattern of shifts will indicate formation or absence of disulfides, and allow conclusions on their positioning. Combinations of alkylation, reduction, alkylation with a different agent followed by mass spectrometry may indeed directly show which sulfhydryls are in a disulfide bond or free. This analysis may also be done prior to and post fusion of cells, which may in fact show the isomerization reaction.

Assigning disulfide bonds to specific cysteines tends to be difficult if more than one pair of disulfide bonds are present in the protein. Additionally, mutational analysis of single cysteine residues in a compact protein as HBsAg with 14 cysteine amino acids can result in artifacts, since it can lead to formation of new disulfide bonds with other unmatched cysteines that do not occur in the wt HBV GPs. Also, we would like to recall that under non-reducing conditions, a previous study (Gallagher et al., 2017) showed that HBsAg analyzed under non-reducing conditions is only observed at the top of the gel, which precludes detection of small electrophoretic shifts by western blot, and that detection of HBsAg monomers or small oligomers was only possible after reduction with DTT.

Importantly, we would like to stress that alkylation of HBV GPs prevents the recognition of the S domain by antibodies targeting epitopes within this region. We invite this Reviewer to read our response to the comment #3 of Reviewer #2. Briefly, we aimed at experimentally proving that amino acids substitutions T303C and G308C induced the formation of an extra disulfide bond by labeling wt or double mutant cysteine LMS with maleimide-containing reagents, i.e., 4-acetamido-40maleimidylstilbene-2,2’-disulfonic acid (AMS) or 2 kDa PEG-maleimide polymer (mPEG) that specifically bind free thiol groups. While HBV GPs were readily detectable after TCEP treatment, alkylation of HBV GPs with AMS prevented their recognition by the Murex antibody that targets the AGL of HBsAg. To circumvent this, we blotted the membranes with an anti-preS1 antibody instead, leading to detection of the L protein only. Treatment with AMS increased the HBV envelope protein’s mass of both wt and TG/CC mutant, though the 1kDa difference between the wt and the TG/CC mutant was not clear. Treatment of HBV GPs with mPEG rendered them undetectable by the two antibodies used.

3) It would be important to demonstrate where ERp57 functions to allow fusion. Is it at the plasma membrane or within virus-containing endosomes? Do they physically interact? Perhaps an isomerization assay would increase the confidence that ERp57 can directly act on the viral protein.

This point is similar to that of Reviewer #1 and we thank both Reviewers for giving us the opportunity to address experimentally this important question. While PDIs are generally abundantly located in the ER, their capacity to traffic beyond the ER in the secretory pathway has been documented and seems to depend both on specific PDI members and cell types. For example, in addition to the ER, ERp57 has been detected at the cell surface as well as in the nucleus, plasma membrane rafts, and cytosol, as discussed in Turano et al. (2002) and references herein. Of note, ERp46, ERp57 and ERp72 were selected in our study because they can be detected at the cell surface. These considerations are now better discussed in the revised version of our manuscript (pages 9 and 13).

Noteworthy, we have experimentally addressed this important concern in our Huh7 cells-based assays.

First, we show in revised Figure 7A a FACS analysis of Huh7-NTCP cells that reveal a significant albeit low expression of ERp57 at the cell surface, which supports our results of cell-cell fusion.

Furthermore, we also show in a new Figure 8 some immuno-fluorescence images by confocal microscopy analysis of Huh7-NTCP cells stained with antibodies against ERp57 and Rab5 (early endosomes), Rab7 (late endosomes), Rab11 (recycling endosomes) or Lamp1 (lysosomes). The quantifications of these results show that ERp57 can be detected in late endosomes but poorly in the other above-tested locations. We believe that these results are meaningful since previous reports showed that HBV infection of HepaRG cells depends on Rab5 and Rab7 (Macovei et al., 2013), which are GTPases involved in the biogenesis of endosomes, and that the epidermal growth factor receptor (EGFr) is a host-entry cofactor that interacts with NTCP and mediates HBV internalization (Iwamoto et al., 2019).

Altogether, we confirm that in addition to the ER, ERp57 can be found at locations compatible for both cell-cell fusion and cell-free entry by internalization. The Results and Discussion sections have been modified accordingly (pages 9-10, pages 13-14).

Finally, as for the question of the Reviewer of the physical interaction between ERp57 and viral particles, we found no difference for virion binding to the cell surface for either NTZ-blocked ERp57expressing (Figure 6A) or ERp57-silenced cells (data not shown). However, these results do not rule out that virus/ERp57 interaction could be weak or transient or, alternatively, could preferentially occurs in late endosomes, as above suggested.

References:

Abou-Jaoudé, G., and Sureau, C. (2007). Entry of hepatitis δ virus requires the conserved cysteine residues of the hepatitis B virus envelope protein antigenic loop and is blocked by inhibitors of thiol-disulfide exchange. J. Virol. 81, 13057–13066.

Earp, L.J., Delos, S.E., Park, H.E., and White, J.M. (2004). The Many Mechanisms of Viral Membrane Fusion Proteins. In Membrane Trafficking in Viral Replication, M. Marsh, ed. (Berlin, Heidelberg: Springer Berlin Heidelberg), pp. 25–66.

Gallagher JR, Torian U, McCraw DM, Harris AK. (2017). Characterization of the disassembly and reassembly of the HBV glycoprotein surface antigen, a pliable nanoparticle vaccine platform. Virology. 502:176-187.

Gudima S, He Y, Meier A, Chang J, Chen R, Jarnik M, Nicolas E, Bruss V, Taylor J. (2007). Assembly of Hepatitis Δ Virus: Particle Characterization, Including the Ability To Infect Primary Human Hepatocytes. Journal of Virology 81:3608-3617.

Huovila AP, Eder AM, Fuller SD. (1992). Hepatitis B surface antigen assembles in a post-ER, pre-Golgi compartment. J Cell Biol. 118(6):1305-20.

Iwamoto, M., Saso, W., Sugiyama, R., Ishii, K., Ohki, M., Nagamori, S., Suzuki, R., Aizaki, H., Ryo, A., Yun, J.-H., et al. (2019). Epidermal growth factor receptor is a host-entry cofactor triggering hepatitis B virus internalization. Proceedings of the National Academy of Sciences 116, 8487– 8492.

Jaoudé GA, Sureau C. Role of the antigenic loop of the hepatitis B virus envelope proteins in infectivity of hepatitis δ virus. J Virol. 2005;79(16):10460-10466. doi:10.1128/JVI.79.16.1046010466.2005.

Julithe R, Abou-Jaoudé G, Sureau C. (2014). Modification of the Hepatitis B Virus Envelope Protein Glycosylation Pattern Interferes with Secretion of Viral Particles, Infectivity, and Susceptibility to Neutralizing Antibodies. Journal of Virology 88:9049-9059.

Lambert C, Prange R. Dual topology of the hepatitis B virus large envelope protein: determinants influencing post-translational pre-S translocation. J Biol Chem. 2001 Jun 22;276(25):22265-72.

doi: 10.1074/jbc.M100956200. Epub 2001 Apr 11. PMID: 11301328.

Lavillette, D., Bartosch, B., Nourrisson, D., Verney, G., Cosset, F.-L., Penin, F., and Pécheur, E.-I. (2006). Hepatitis C virus glycoproteins mediate low pH-dependent membrane fusion with liposomes. J. Biol. Chem. 281, 3909–3917.

Le Duff Y, Blanchet M, Sureau C. The pre-S1 and antigenic loop infectivity determinants of the hepatitis B virus envelope proteins are functionally independent. J Virol. 2009 Dec;83(23):12443-51. doi: 10.1128/JVI.01594-09. Epub 2009 Sep 16. PMID: 19759159; PMCID: PMC2786703.

Lin, A.H., and Cannon, P.M. (2002). Use of pseudotyped retroviral vectors to analyze the receptor binding pocket of hemagglutinin from a pathogenic avian influenza A virus (H7 subtype). Virus Res 83, 43–56.

Macovei, A., Petrareanu, C., Lazar, C., Florian, P., and Branza-Nichita, N. (2013). Regulation of hepatitis B virus infection by Rab5, Rab7, and the endolysosomal compartment. J. Virol. 87, 6415–6427.

Mangold CM, Unckell F, Werr M, Streeck RE. Secretion and antigenicity of hepatitis B virus small envelope proteins lacking cysteines in the major antigenic region. Virology. 1995 Aug 20;211(2):535-43. doi: 10.1006/viro.1995.1435. PMID: 7645257.

Perez-Vargas, J., Amirache, F., Boson, B., Mialon, C., Freitas, N., Sureau, C., Fusil, F., and Cosset, F.-L. (2019). Enveloped viruses distinct from HBV induce dissemination of hepatitis D virus in vivo. Nature Communications 10.

Richard, C., Liuzzo, J.P., and Moscatelli, D. (1995). Fibroblast Growth Factor-2 Can Mediate Cell Attachment by Linking Receptors and Heparan Sulfate Proteoglycans on Neighboring Cells. Journal of Biological Chemistry 270, 24188–24196.

Salisse J, Sureau C. A function essential to viral entry underlies the hepatitis B virus "a" determinant. J Virol. 2009 Sep;83(18):9321-8. doi: 10.1128/JVI.00678-09. Epub 2009 Jul 1. PMID: 19570861; PMCID: PMC2738268.

Schulze, A., Gripon, P., and Urban, S. (2007). Hepatitis B virus infection initiates with a large surface protein-dependent binding to heparan sulfate proteoglycans. Hepatology 46, 1759–1768.

Suffner S, Gerstenberg N, Patra M, Ruibal P, Orabi A, Schindler M, Bruss V. (2018). Domains of the Hepatitis B Virus Small Surface Protein S Mediating Oligomerization. J Virol. 92(11):e02232-17.

Sureau C, Salisse J. A conformational heparan sulfate binding site essential to infectivity overlaps with the conserved hepatitis B virus a-determinant. Hepatology. 2013 Mar;57(3):985-94. doi: 10.1002/hep.26125. Epub 2013 Feb 15. PMID: 23161433.

Turano, C., Coppari, S., Altieri, F., and Ferraro, A. (2002). Proteins of the PDI family: unpredicted nonER locations and functions. J. Cell. Physiol. 193, 154–163.

Vyas GN, Rao KR, Ibrahim AB. Australia antigen (hepatitis B antigen): a conformational antigen dependent on disulfide bonds. Science. 1972 Dec 22;178(4067):1300-1. doi: 10.1126/science.178.4067.1300. PMID: 4118259.